# The thalamus encodes and updates context representations during hierarchical cognitive control

**Xitong Chen**[1,2,3]*, **Stephanie C. Leach**[1,2,3], **Juniper Hollis**[1,2,3], **Dillan Cellier**[1,2,3], **Kai Hwang**[1,2,3]

**1** Department of Psychological and Brain Sciences, The University of Iowa, Iowa City, Iowa, United States of America, **2** Cognitive Control Collaborative, The University of Iowa, Iowa City, Iowa, United States of America, **3** Iowa Neuroscience Institute, The University of Iowa, Iowa City, Iowa, United States of America

\* xitong-chen@uiowa.edu

**Data Availability Statement:** The numerical data supporting this study are available in the supplementary file S1 Data. The neuroimaging data have been deposited in NeuroVault and can be

## Abstract

Cognitive flexibility relies on hierarchically structured task representations that organize task contexts, relevant environmental features, and subordinate decisions. Despite ongoing interest in the human thalamus, its role in cognitive control has been understudied. This study explored thalamic representation and thalamocortical interactions that contribute to hierarchical cognitive control in humans. We found that several thalamic nuclei, including the anterior, mediodorsal, ventrolateral, and pulvinar nuclei, exhibited stronger evoked responses when subjects switch between task contexts. Decoding analysis revealed that thalamic activity encodes task contexts within the hierarchical task representations. To determine how thalamocortical interactions contribute to task representations, we developed a thalamocortical functional interaction model to predict task-related cortical representation. This data-driven model outperformed comparison models, particularly in predicting activity patterns in cortical regions that encode context representations. Collectively, our findings highlight the significant contribution of thalamic activity and thalamocortical interactions for contextually guided hierarchical cognitive control.

## Introduction

Cognitive control enables humans to adjust their behaviors flexibly in response to changing circumstances. For example, the sound of a ringing phone can elicit different reactions depending on the context. At home, one might answer, while driving might cause one to ignore it for safety. Everyday life is filled with scenarios that require us to modify our actions based on different contexts. How does our control system facilitate such contextually guided control?

One influential framework posits that cognitive control is supported by hierarchically organized task representations that are encoded by the functional organization of the prefrontal cortex [1–3]. In this framework, task representations are structured to encode contexts, goals, task-relevant features, and potential actions [4]. Furthermore, representations can be

accessed at https://identifiers.org/neurovault.
collection:18728. All raw data have been made
publicly accessible on OpenNeuro (https://
openneuro.org/datasets/ds005600/). Additionally,
the analysis code used in this study has been
archived on GitHub (https://github.com/
HwangLabNeuroCogDynamics/Thalamocortical_
Hierarchical_Control), with a DOI generated via
Zenodo (DOI: 10.5281/zenodo.14086485).

**Funding:** Research reported in this publication was
supported by the Iowa Neuroscience Institute
(https://medicine.uiowa.edu/iowaneuroscience)
and the National Institute of Mental Health (https://
www.nimh.nih.gov) under Award Number
R01MH122613, awarded to K.H. This work was
also conducted on an MRI instrument funded by
the National Institutes of Health (https://www.nih.
gov) under grant number 1S10OD025025-01. The
funders had no role in study design, data collection
and analysis, decision to publish, or preparation of
the manuscript.

**Competing interests:** The authors have declared
that no competing interests exist.

**Abbreviations:** BOLD, blood oxygenated level-
dependent; CM, center of mass; CR, cue repeat;
CS, cue switch; CSF, cerebral spinal fluid; DLPFC,
dorsal lateral prefrontal cortex; EDS, extra-
dimension switch; FC, functional connectivity; FD,
framewise displacement; FDR, false discovery rate;
fMRI, functional magnetic resonance imaging;
FWER, family-wise error rate; GLM, general linear
model; HRF, hemodynamic response function;
IDS, inter-dimension switch; MVPA, multivoxel
pattern analysis; PCA, principal component
analysis; PFC, prefrontal cortex; REML, restricted
maximum likelihood; RSA, representational
similarity analysis; RT, response time; WM, white
matter.

organized in a hierarchical manner, where more abstract contextual representations exert stronger influence over action representations to support cognitive flexibility [5,6]. Anatomically, it was proposed that this abstraction gradient is supported by a rostral to caudal organization in the lateral prefrontal cortex [2,7–10].

To support flexibility, task representations are updated as context changes. One computation model suggests that the cortico-striatal-thalamic circuit is involved in updating cortical task representations [3,11,12]. According to this model, the basal ganglia receives control signals from the prefrontal cortex, and then uses these signals to disinhibit the thalamus to influence cortical representations via thalamocortical inputs [3,11]. Inspired by this model, several neuroimaging studies focused on characterizing the interaction between prefrontal cortex and the striatum [13–15].

Notably, the human thalamus is a critical but understudied component of this circuitry. Structurally, the basal ganglia does not directly project to the cortex, therefore its influence on cortical representations must be mediated by the thalamus [16–18]. Despite this anatomical relationship, many theoretical models omit it from the model. Furthermore, most existing models are primarily built on the known projections from the pallidum to the ventrolateral thalamus for allowing motoric representations to be influenced by higher-order control processes [12,19–22]. It remains unclear whether higher order thalamic nuclei, including the anterior, mediodorsal, and pulvinar nuclei, which have dense connectivity with frontal and parietal regions [23–25], are involved in hierarchical cognitive control in humans. This is because most past neuroimaging studies did not provide specific details on the thalamic functional anatomy. Furthermore, recent findings from animal models demonstrated that thalamocortical interactions can signal context changes [26–28]. In humans, the thalamus is ideally positioned to influence cortical representations via its extensive one-to-many and many-to-one thalamocortical connectivity motifs [29]. Specifically, the anterior, medial, and posterior thalamus exhibit strong converging connectivity with multiple frontal-parietal systems that have been implicated in control-related functions [30]. The behavioral significance of this thalamocortical connectivity profile is further affirmed by recent functional neuroimaging studies [31,32] and lesion evidence [32,33]. However, it remains unclear how thalamocortical connectivity supports hierarchical cognitive control.

The current study focused on elucidating the role of the human thalamus in hierarchical cognitive control. We designed a paradigm with different levels of hierarchical task representations. Instead of using different tasks to establish hierarchy, we equated stimuli and rule complexity across different levels of hierarchical representations within one behavioral task design [5]. We had 3 main objectives. First, we aimed to characterize the detailed functional anatomy of thalamic activity that supports hierarchical cognitive control. Second, we aimed to determine the interaction between thalamic activity and cortical control representations. To accomplish this, we utilized a novel, data-driven, thalamocortical activity flow mapping analysis to test if thalamocortical interactions can predict cortical activity patterns related to cognitive control [31,33–35]. Third, we decoded the cognitive representations encoded by thalamic activity to determine the hierarchical task representation to which thalamocortical interactions most strongly contribute.

## Results

### Hierarchical control task and behavioral results

We designed a behavioral task to investigate hierarchical cognitive control (Fig 1A). In each trial, a task cue was presented for 0.5 second, followed by a picture probe for 2.5 seconds. Subjects had to respond to the probe by applying one of 2 specific response rules: the "scene rule"

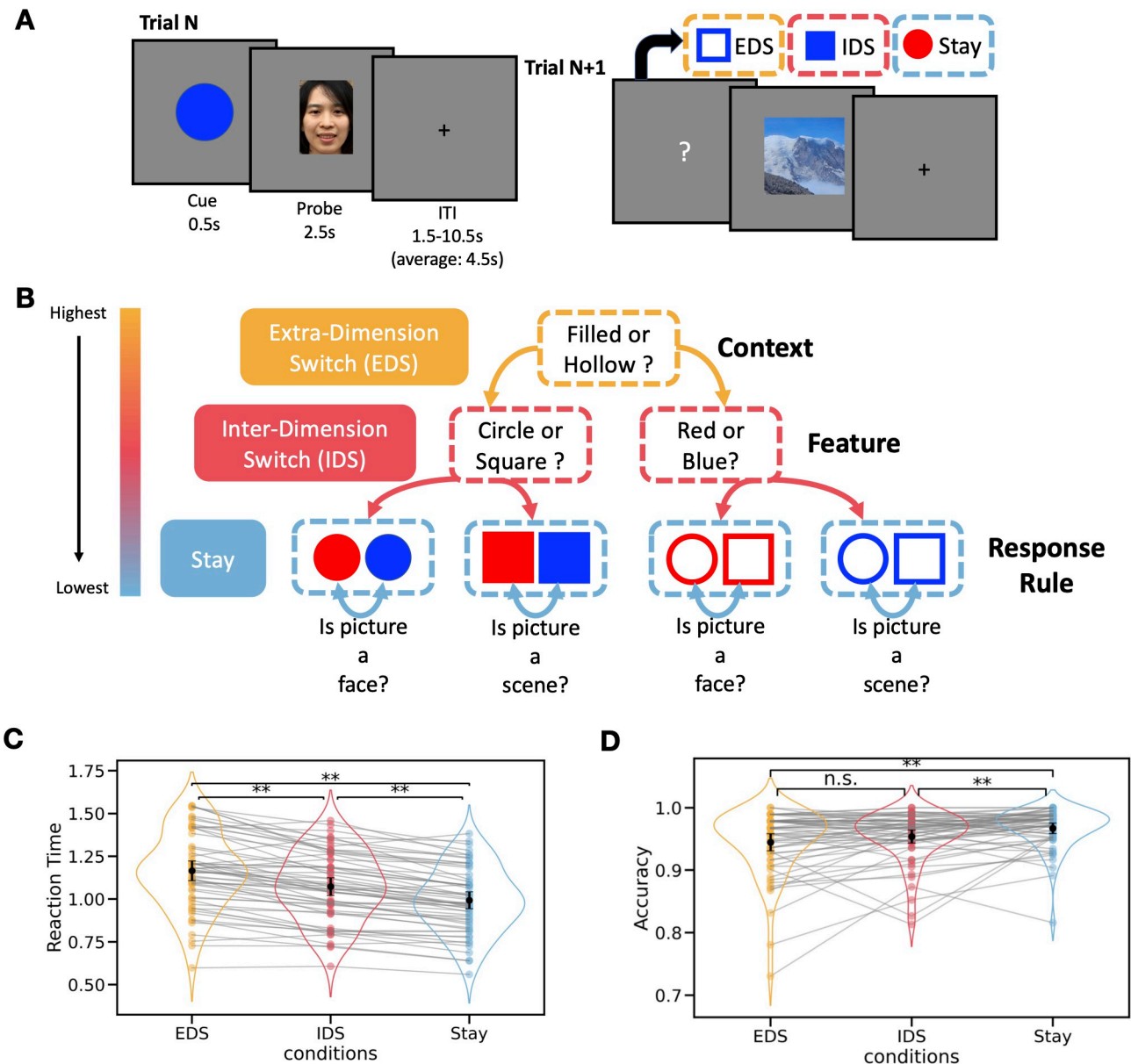

**Fig 1. Hierarchical control task.** (A) The hierarchical control task, in which we introduced different levels of hierarchical task switching between trials. (B) The hierarchy of task representations. (C) Reaction time observed across the 3 hierarchical levels of task-switching. (D) Accuracy across the 3 hierarchical levels of task-switching. ** $p < 0.01$. n.s., nonsignificant. The error bar represents the 95% confidential interval. The black dot indicates the mean value, while the colored dot represents data from individual subjects. Lines connect the data points for each subject across different conditions. Data used for (C) and (D) can be found in S1 Data, specifically in the sheet labeled "Fig 1C and 1D".

(determining whether the presented picture was a scene) or the "face rule" (determining whether the presented picture was a human face). The appropriate rule selection was based on different features of the cue, organized hierarchically. Specifically, the texture of the cue (solid versus hollow) referred to as the "context," representing the highest level of attributes. This context information determined which mid-level "feature" subjects should select to determine the appropriate "response rule." Each feature from the mid-level (shape or color) was then associated with either the face or scene rule. The face and scene rule then determines the visual stimuli category subjects should decide to attend to generate the correct motor response. This

organization introduced three hierarchical levels of task switching: extra-dimension switch (EDS), inter-dimension switch (IDS), and Stay (Fig 1B). In EDS trials, subjects updated the highest-level task context. In IDS trials, subjects switched between task-relevant features within the same task context. In Stay trials, subjects performed the same task for 2 trials in a row. We hypothesized that as the hierarchical task switching demand increases from Stay to IDS to EDS, response time (RT) would also increase.

Results from the one-way (Conditions: EDS, IDS, and Stay) repeated measures ANOVA (rmANOVA) showed that the hypothesized hierarchical structure of task representations significantly modulated subject's RT ($F(2,116) = 156.21$, $p < 0.0001$, $\eta^2 = 0.11$, Fig 1C). Subjects responded slowest for the EDS condition (mean ± SD: 1.16 s ± 0.22; EDS versus IDS: $t(58) = 8.36$, $p < 0.001$, Cohen's d = 0.44; EDS versus Stay: $t(58) = 16.70$, $p < 0.001$, Cohen's d = 0.83), followed by the IDS (mean ± SD: 1.07 s ± 0.20; IDS versus Stay: $t(58) = 10.60$, $p < 0.001$, Cohen's d = 0.41) and Stay (mean ± SD: 0.99 s ± 0.19) conditions. Similar effects were observed for accuracy ($F(2,116) = 13.70$, $p < 0.0001$, $\eta^2 = 0.05$; Fig 1D). Accuracy for the EDS condition (mean ± SD: 0.94 ± 0.05) was significantly lower than the Stay condition (mean ± SD: 0.97 ± 0.03; $t(58) = -6.10$, $p < 0.001$, Cohen's d = −0.51), but not significantly different to the IDS condition (mean ± SD: 0.95 ± 0.04; $t(58) = -1.88$, $p = 0.19$, Cohen's d = −0.20). The accuracy for the IDS condition was significantly lower than Stay ($t(58) = -3.12$, $p = 0.008$; Cohen's d = −0.36). We next determined whether these effects were affected by repeating responses (choose "yes" or "no") in the EDS and IDS conditions. Significant differences were found between EDS and IDS for both response switch (EDS versus IDS: $t(58) = 6.61$, $p < 0.0001$, Cohen's d = 0.43; EDS versus Stay: $t(58) = 15.01$, $p < 0.0001$, Cohen's d = 0.95; IDS versus Stay: $t(58) = 11.93$, $p < 0.0001$, Cohen's d = 0.56) and response repeat (EDS versus IDS: $t(58) = 8.14$, $p < 0.0001$, Cohen's d = 0.44; EDS versus Stay: $t(58) = 13.38$, $p < 0.0001$, Cohen's d = 0.70; IDS versus Stay: $t(58) = 6.08$, $p < 0.0001$, Cohen's d = 0.25; S1 Fig). We then examined whether task-switching effects were influenced by repeating cues in the Stay condition. Significant task-switching effects were observed when comparing both cue repeat (CR) and cue switch (CS) Stay trials for RT and accuracy. For RT, participants were significantly slower in the EDS condition compared to both Stay_CS ($t(58) = 12.99$, $p < 0.0001$, Cohen's d = 0.62) and Stay_CR ($t(58) = 16.80$, $p < 0.0001$, Cohen's d = 0.97; S1 Fig) trials. Similarly, participants were slower in the IDS condition compared to Stay_CS ($t(58) = 4.31$, $p = 0.001$, Cohen's d = 0.19) and Stay_CR ($t(58) = 12.32$, $p < 0.0001$, Cohen's d = 0.55). Additionally, participants responded more slowly in Stay_CS trials compared to Stay_CR trials ($t(58) = 7.42$, $p < 0.0001$, Cohen's d = 0.36; S1 Fig). For accuracy, performance in the EDS condition was significantly lower than in Stay_CS ($t(58) = -3.97$, $p = 0.002$, Cohen's d = 0.30) and Stay_CR ($t(58) = -8.01$, $p < 0.0001$, Cohen's d = 0.70; S1 Fig) trials. In the IDS condition, accuracy was not significantly different from Stay_CS ($t(58) = -0.125$, $p = 1.00$, Cohen's d = 0.01) but was significantly lower than in Stay_CR ($t(58) = -4.94$, $p < 0.0001$, Cohen's d = 0.45). Furthermore, accuracy in Stay_CS trials was significantly lower than in Stay_CR trials ($t(58) = -5.30$, $p < 0.0001$, Cohen's d = 0.46; S1 Fig). This observed pattern of reaction time and accuracy results suggest a hierarchical organization of task representations, consistent with findings from our previous published study that employed the same task with EEG [5]. Furthermore, these behavioral results pattern remained consistent when separated trials where the response and cue in the current trial was repeated from the previous trial.

## Cortical evoked responses

We found that the behavioral performance was influenced by the demand of hierarchical task switching. Subsequently, we explored how hierarchical task switching modulated cortical and

subcortical activity. We first investigated cortical evoked responses elicited by different hierarchical task switching conditions. This analysis identified a set of cortical regions including the middle frontal gyrus, the inferior frontal sulcus, the precentral sulcus, the intraparietal sulcus, the postcentral sulcus, the insula, the medial frontal gyrus, the posterior cingulate gyrus, the lateral occipital gyrus, the fusiform gyrus, the posterior cingulate cortex, and the inferior temporal gyrus (EDS, IDS, and Stay conditions; Fig 2A; center of mass (CM) coordinates available in Table A, B, and C in S1 Text). We then contrasted the magnitudes of evoked responses between different task switching conditions (e.g., EDS-Stay, EDS-IDS, and IDS-Stay; Fig 2B). Specifically, we examined EDS versus Stay and EDS versus IDS to determine cortical regions involved in switching of task context, and IDS versus Stay for switching of task feature. We found stronger evoked responses in the frontoparietal and the temporal regions in response to changing task contexts (EDS-Stay and EDS-IDS, Fig 2B; CM coordinates available in Table D, E and F in S1 Text), included areas such as the rostral middle frontal gyrus, the inferior frontal sulcus, the insula, the precentral sulcus, the postcentral sulcus, the intraparietal sulcus, the medial frontal gyrus, the posterior cingulate gyrus, the precuneus, the cuneus, and the middle temporal lobe. We did not observe significant effects in these regions during switches in feature level task information (IDS-Stay). Instead, when contrasting task switching that involved feature level task information (IDS--Stay), we found significant clusters primarily distributed in the premotor cortex and the intraparietal sulcus. These cortical regions also showed significant modulations in response to updating context information (EDS-IDS). This pattern revealed an asymmetrical hierarchy consistent with previous studies [2], in which several regions in the rostral frontal cortex are more involved in contextual update, whereas premotor regions are responsive to both contextual and feature level updates of task representations. The overall cortical response pattern remained consistent when considering trials where the current trial's response (choose "yes" or "no") and cue were repeated from the previous trial (see S2 Fig).

## Thalamic activity profile responding to hierarchical task-switching

We then investigated the detailed functional anatomy of thalamic activity in response to different levels of hierarchical task switching (EDS, IDS, and Stay conditions; unthresholded activity maps: Fig 3A, thresholded activity maps at cluster corrected $p < 0.05$: Fig 3B). Significant evoked responses were observed in the anterior, the ventral, the medial, and the posterior regions of the thalamus for all task switching conditions (Fig 3A and 3B). The negative response patterns observed in the medial and the posterior region of the thalamus may be attributed to a significant undershoot in the hemodynamic response functions (HRFs) of these nuclei (see S3 Fig).

Our results revealed activity patterns within the thalamus for updating different hierarchical task representations. Thalamic voxels in the anterior, the ventroanterior, the ventromedial, the ventrolateral, the mediodorsal, and the intralaminar nuclei showed stronger evoked responses that were more selective for updating context representations (EDS-IDS; Fig 3C). In addition, we observed that voxels in the anterior, the ventroanterior, the ventrolateral, the mediodorsal, the intralaminar, the medial pulvinar, the lateral pulvinar, and the ventral posterior nuclei of the thalamus showed stronger evoked responses for the EDS-Stay contrast (Fig 3C). We did not observe any significant clusters for the IDS-Stay contrast, but only for the EDS-IDS and EDS-Stay contrasts. This suggests that the thalamus may be more selectively involved in hierarchical cognitive control that involves updating context representations. This overall thalamic response pattern remained consistent when considering trials where the current trial's response and cue were repeated from the previous trial (see S4 Fig).

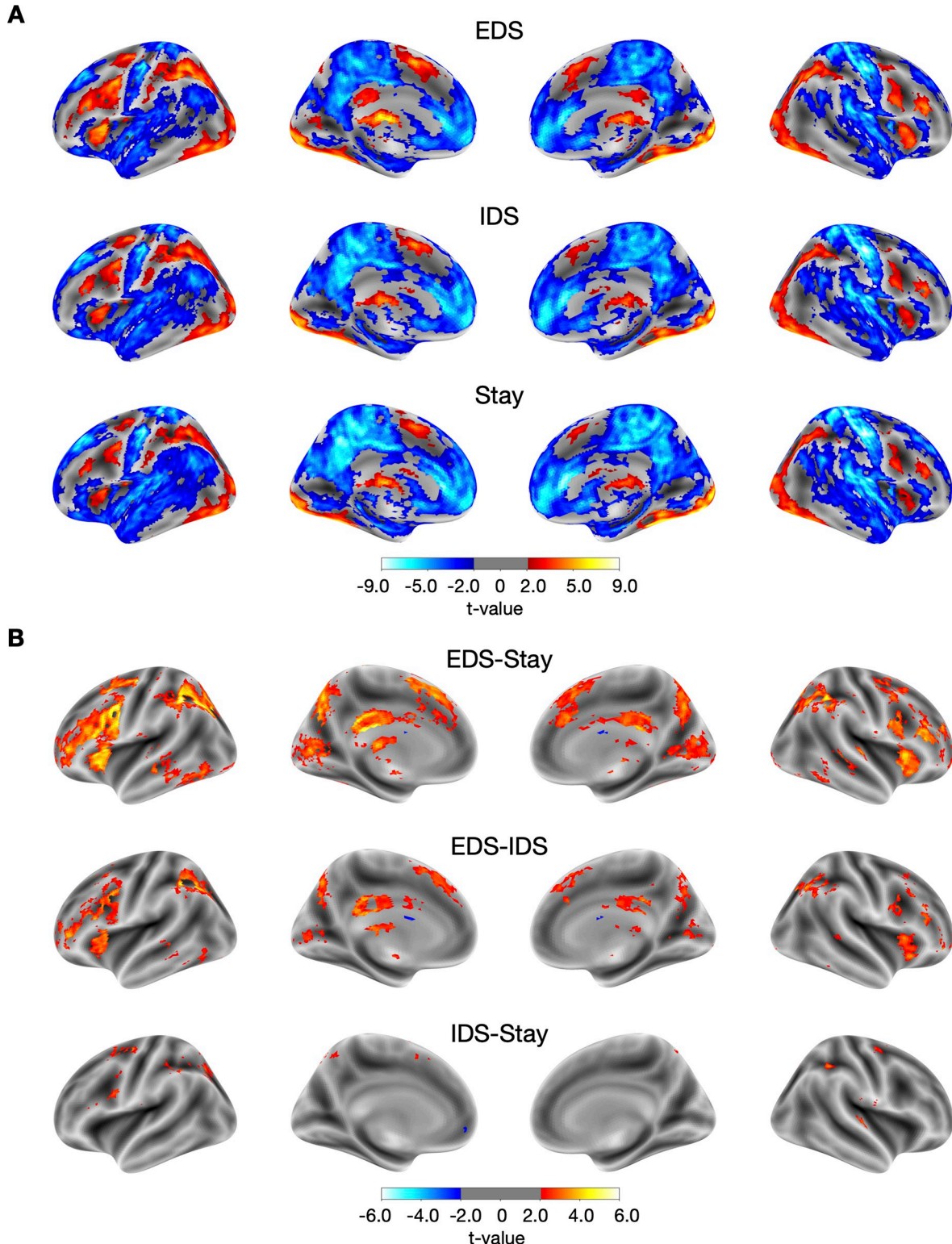

**Fig 2. Cortical evoked response to hierarchical task-switching.** (A) Cortical evoked responses for the three task-switching conditions. (B) Contrast between conditions. The results were first thresholded at a voxel-level threshold of $p < 0.05$, followed by cluster correction procedure with a cluster level threshold of $p < 0.05$, only showing clusters with a minimum cluster size (k) of 58 voxels. Clusters were defined as groups of voxels that are connected by sharing a face with their neighboring voxels. The group statistical maps presented in Fig 2 can be accessed at https://identifiers.org/neurovault.collection:18728.

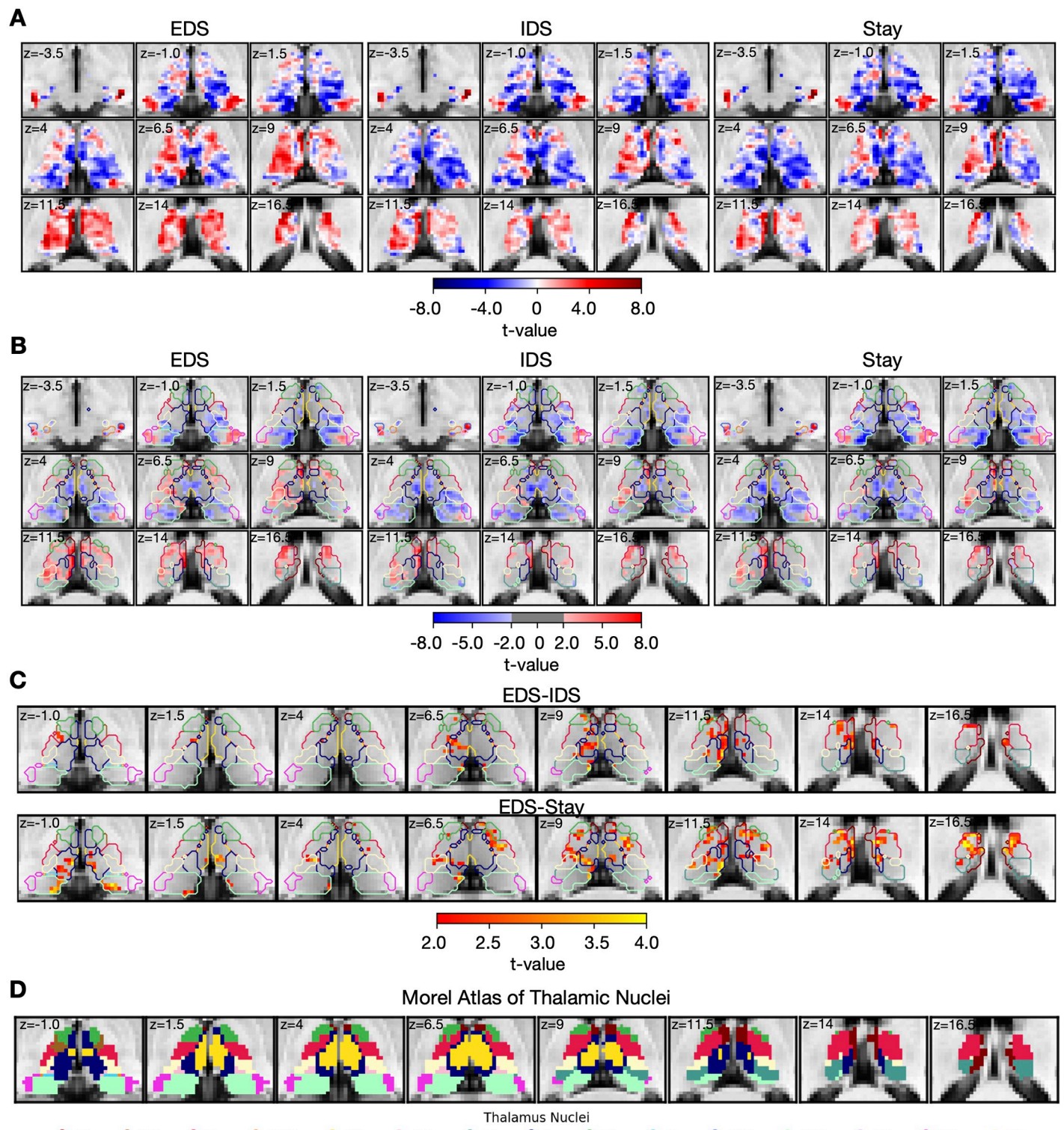

**Fig 3. Thalamic voxel-wise evoked responses in response to hierarchical task-switching.** (A) Unthresholded thalamic evoked response patterns (t-value) for the 3 hierarchical task-switching conditions. (B) Thresholded thalamic evoked response patterns. (C) Contrast between conditions. The results were first thresholded at a voxel-level threshold of $p < 0.05$, followed by cluster correction procedure with a cluster level threshold of $p < 0.05$, only showing clusters with a minimum cluster size (k) of 58 voxels. Clusters were defined as groups of voxels that are connected by sharing a face with their neighboring voxels. The color of contour in (B) and (C) refers

to the nuclei described in (D). (D) Morel atlas of thalamic nuclei. Abbreviations for thalamic nuclei are as follows: AN, anterior nucleus; VM, ventral medial nucleus; VL, ventral lateral nucleus; MGN, medial geniculate nucleus; MD, mediodorsal nucleus; PuA, anterior pulvinar nucleus; LP, lateral posterior nucleus; IL, intralaminar nucleus; VA, ventral anterior nucleus; Po, posterior nucleus; LGN, lateral geniculate nucleus; PuM, medial pulvinar nucleus; PuL, lateral pulvinar nucleus; VP, ventral posterior nucleus. Similar results were also observed using an 8 mm smoothing kernel, voxel-level threshold of $p < 0.001$, and cluster threshold of $p < 0.05$ (cluster size of 42 voxels), see S5 Fig. The group statistical maps presented in Fig 3 can be accessed at https://identifiers.org/neurovault.collection:18728.

In summary, the hierarchical cognitive control modulates the evoked response of thalamus. Additionally, thalamic nuclei in anterior, ventral, and medial regions displayed a preference of updating context-level task representation in hierarchical cognitive control task.

## Thalamic voxels selectively encode contextual information

Our results showed that thalamic activity is most strongly associated with updating contextual information during hierarchical cognitive control. We next seek to determine which level of hierarchical task representation is encoded in the thalamus. Utilizing multivoxel pattern analysis, we investigated whether we could decode context, color, shape, or task from multivoxel thalamic activity patterns. We observed above chance decoding accuracy of contextual information throughout the thalamus after applying a stringent cluster correction threshold of $p < 0.001$ (solid versus hollow; Fig 4). Highest decoding accuracy was found in the ventroanterior ($0.52 \pm 0.004$), the ventrolateral ($0.52 \pm 0.004$), the mediodorsal ($0.52 \pm 0.004$), the intralaminar ($0.52 \pm 0.004$), the medial pulvinar ($0.52 \pm 0.004$), and the lateral posterior nuclei ($0.52 \pm 0.003$). These thalamic nuclei were also found to show strongest evoked response during the contextual update condition (EDS). Decoding performance for other information, color (blue versus red), shape (square versus circle), and task (face versus scene tasks), was significantly weaker, with few significant voxels in anterior and medial regions (Fig 4). But these voxels did not survive after cluster-correction. These results suggest that thalamic activity is involved in encoding contextual information during hierarchical cognitive control.

## Thalamocortical interaction model predicts cortical activity patterns

Given the strong reciprocal and nonreciprocal connections between the thalamus and the cortex, significant thalamocortical interactions may take place during hierarchical cognitive control. Prior studies using computational modeling [26] and animal models [27,28] have suggested that thalamocortical interactions instantiate cortical task representations. Therefore, we further examined how thalamocortical interactions in humans contribute to hierarchical

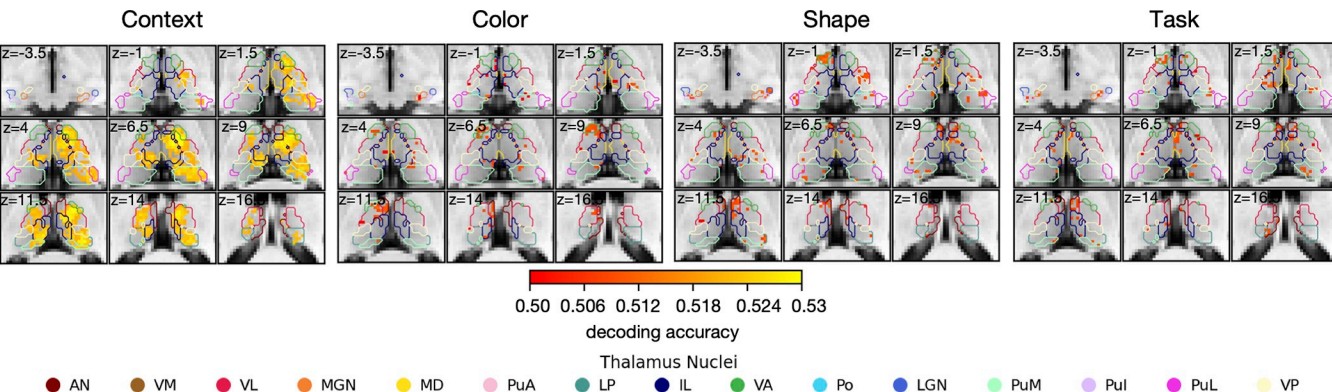

**Fig 4. Thalamic activity encodes task contexts.** Cluster-corrected decoding performance on Context at $p < 0.001$ (minimum cluster size = 343 voxels). Color, shape, and task were uncorrected. The group statistical maps presented in Fig 4 can be accessed at https://identifiers.org/neurovault.collection:18728.

task representations that support cognitive flexibility. To this end, we modified a previously developed activity flow mapping procedure to determine the relationship between thalamic evoked responses, thalamocortical functional connectivity (FC), and cortical representations [34–36]. In brief, this method tests whether a data-driven thalamocortical model can predict cortical activity patterns by weighting evoked response patterns of thalamic voxels based on their FC with cortical ROIs (see Materials and methods for details). This approach allows us to assess the extent to which thalamocortical interactions influence cortical task activity patterns associated with hierarchical task representations.

We first tested whether this thalamocortical model can predict whole-brain activity patterns associated with different hierarchical task-switching conditions (Fig 5A). Briefly, we calculated the dot product of voxel-wise evoked response patterns within the Morel's thalamus mask [37] and their FC matrix between thalamic voxels and 400 ROIs defined by the Schaefer's cortical parcellation [38]. Thalamocortical FC was calculated using principal component regression, which evaluates the FC between thalamus and cortices after regressing out stimuli-evoked signals, a measure also known as "background connectivity" [39]. We employed a split-half cross-validation procedure to generate predicted activity patterns for the 400 ROIs (see Materials and methods for details). To evaluate prediction accuracy, we compared these predicted patterns to observed cortical activity patterns, using Pearson correlation normalized by noise ceiling (see Materials and methods for details). We then compared the performance of our model against 2 null models. The first null model assumed that no spatial pattern of thalamic evoked response carried task-related information. This was tested by randomly shuffling each subject's thalamic evoked response vector with 5,000 random permutations. The second null model assumed that no thalamocortical FC pattern carried task-related information. This was tested by randomly shuffling elements within the FC matrix with 5,000 random permutations. The results demonstrated that our model significantly outperformed the 2 null models across all conditions (Fig 5B and 5C). However, there was no statistically significant difference between the 3 conditions ($F(2,116) = 1.43$, $p = 0.24$; EDS (mean ± SD): 0.50 ± 0.25, IDS (mean ± SD): 0.46 ± 0.20, Stay (mean ± SD): 0.49 ± 0.18).

The above analysis included all thalamic voxels. To investigate the specific contributions of different thalamic subregions to model prediction, we simulated thalamic lesions by systematically removing 20% of thalamic voxels, which we referred to as "virtual lesioning voxels," from the activity flow mapping analysis. These voxels were selected based on their ranking by the amplitudes of evoked responses, ranging from the 1st to the 80th percentile. Our findings consistently showed that the removal of voxels with the strongest evoked responses resulted in a notable decrease in prediction accuracy across all 3 hierarchical task-switching conditions (Fig 5D). After averaging the reduction magnitude across thalamic nuclei, we found that thalamic voxels contributing most significantly to prediction accuracy were primarily in the anterior, the ventrolateral, the mediodorsal, the intralaminar, the ventral posterior, and the medial pulvinar nuclei (Fig 5E and 5F). Notably, these thalamic voxels overlapped with those showing strong evoked response modulations (EDS-Stay and EDS-IDS; Fig 3C). In summary, the above results demonstrated the utility of thalamocortical activity flow prediction. Having demonstrated the utility of this approach, next we built thalamocortical models to predict voxel-wise activity patterns within specific cortical ROIs that encode different task representations.

## Thalamocortical interactions update cortical control representations

Our results from the thalamocortical activity flow analysis so far have demonstrated that thalamocortical interactions can predict cortical activity patterns. However, considering the hierarchical organization of task representations, it remains unclear which specific level of

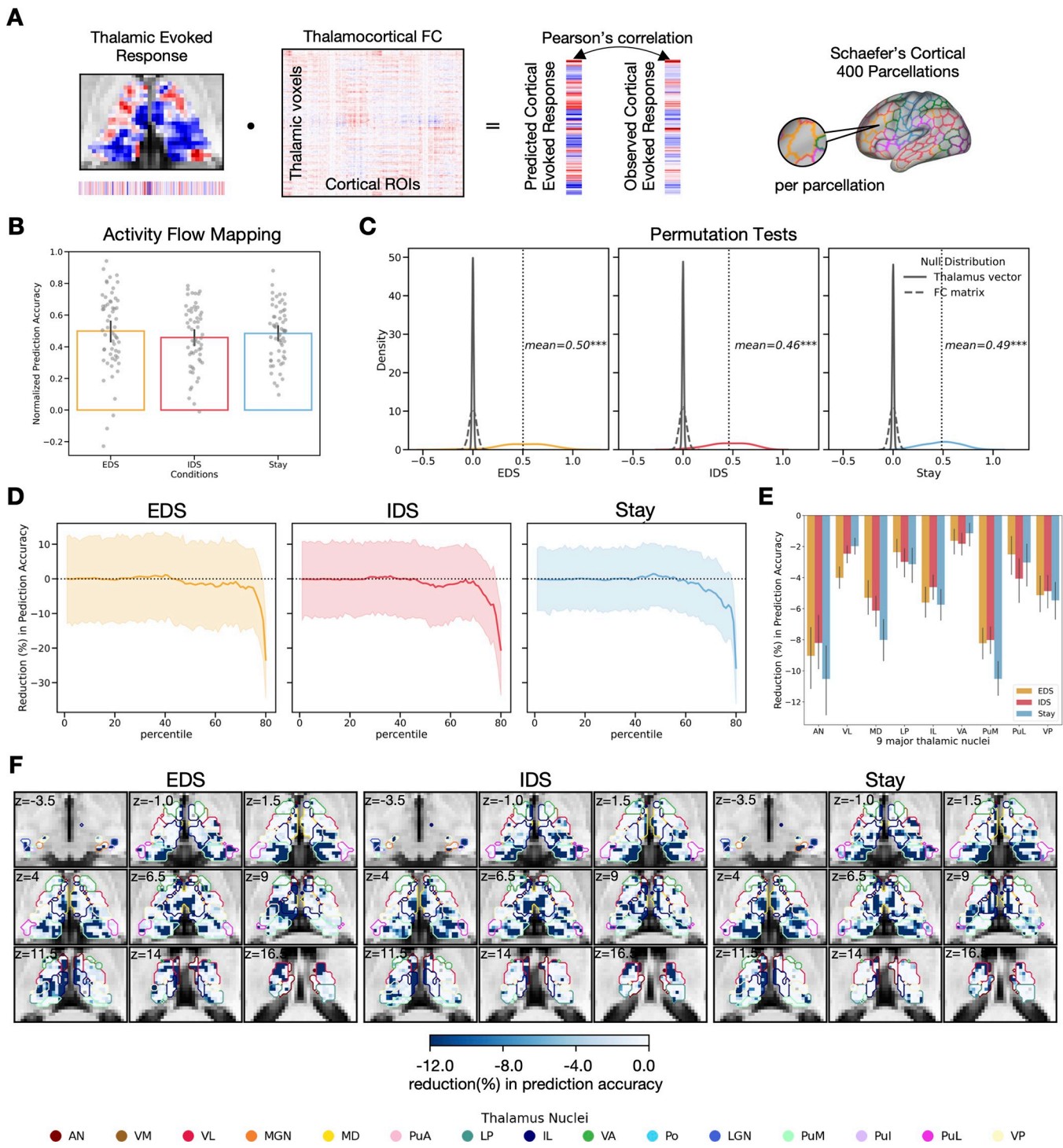

**Fig 5. Thalamocortical activity flow mapping.** (A) Diagram of activity flow mapping analysis, adapted from [34,35]. The thalamus and brain parcellation masks that were used to extract the evoked response and time series data for all activity flow mapping analysis in current study. Schaefer's 400 cortical parcellations [38] were used to calculate voxel-to-parcellation thalamocortical activity flow. (B) The prediction accuracy of voxel-to-ROI thalamocortical activity flow for the hierarchical task-switching conditions. The error bar represents the 95% confidential interval. (C) Null models. (D) Reduction in prediction accuracy for the 3 hierarchical task-switching conditions. (E) Averaged reduction magnitude in prediction accuracy in 9 major thalamic nuclei. (F) Thalamic topography of reduction in prediction accuracy. Data used for (B–E) can be found in S1 Data, specifically in the sheets labeled "Fig 5B," "Fig 5C," "Fig 5D," and "Fig 5E." The group statistical maps presented in (F) can be accessed at https://identifiers.org/neurovault.collection:18728.

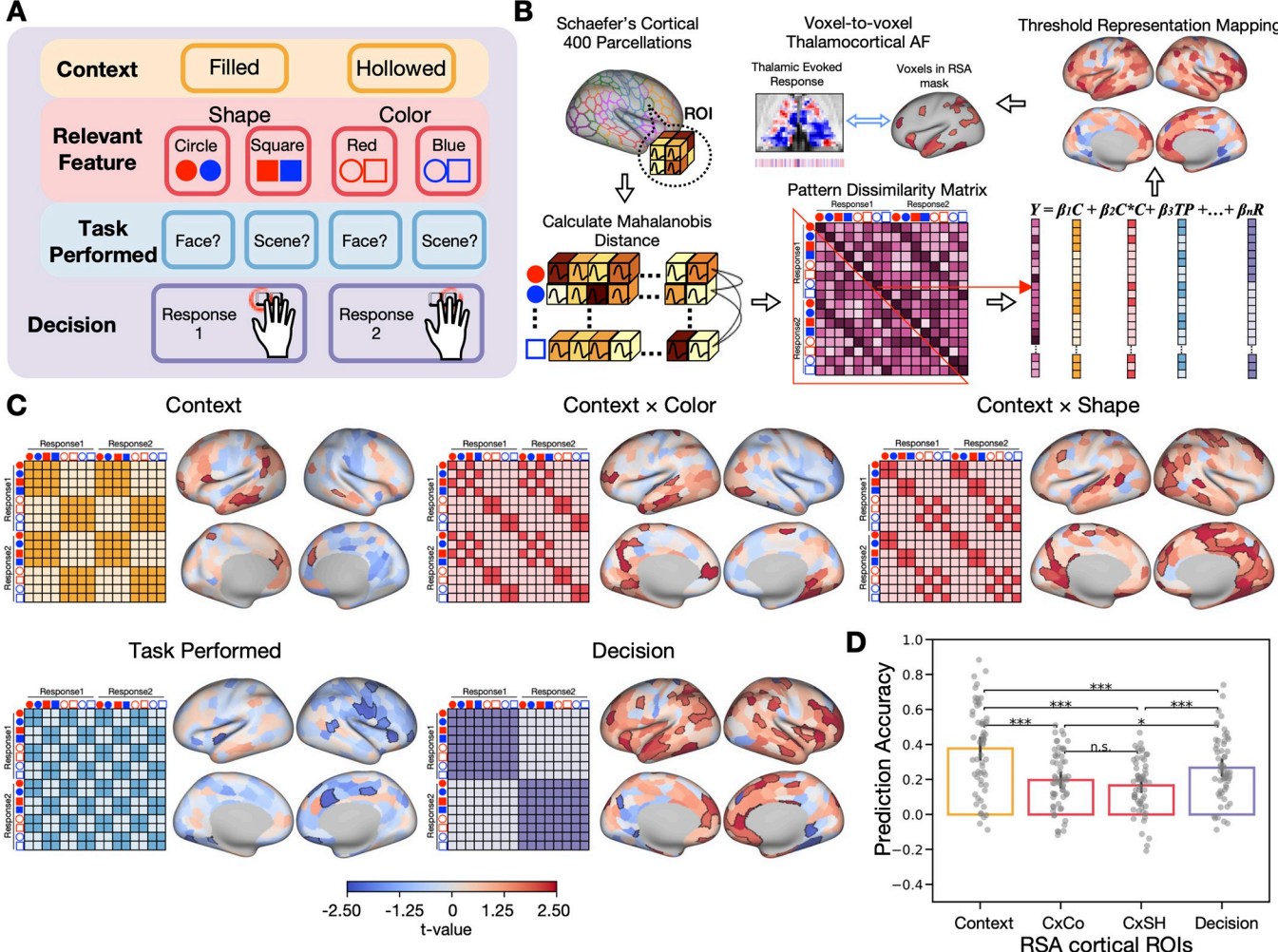

**Fig 6. Thalamocortical updates the cortical control representation.** (A) The hierarchical structure of task representations. (B) Activity flow mapping and ROI-RSA procedures. Within each ROI, we calculated the Mahalanobis distance of cortical activity pattern between different types of cues to evaluate the neural dissimilarity matrix. The lower triangle of the neural dissimilarity matrix was extracted and regressed onto an intercept and different representational models. The resulted beta coefficients were subjected to group-level *t* test. The thresholded results were used as mask for the further voxel-to-voxel thalamocortical interaction model. (C) Five pattern dissimilarity matrix that modeled different representations. Cortical regions that encode different representations within the hierarchical task representation. The white outline in each cortical map depicts statistically significant clusters that used in the voxel-to-voxel thalamocortical activity flow mapping. (D) The prediction accuracy of voxel-to-voxel thalamocortical activity flow model for different hierarchical task representations. Multiple comparisons were corrected by the Bonferroni procedure. * $p < 0.05$; ** $p < 0.01$; *** $p < 0.001$. Data used for (D) can be found in S1 Data, specifically in the sheet labeled "Fig 6D." The group statistical maps presented in (C) can be accessed at https://identifiers.org/neurovault. collection:18728.

representations is influenced by thalamocortical interactions. To address this question, we first conducted a whole-brain representational similarity analysis (RSA) to identify the specific representation encoded by each cortical region. Subsequently, we examined the pattern of thalamocortical interactions with each cortical region using the previously mentioned thalamocortical activity flow model.

We first performed an ROI RSA (Fig 6A and 6B) to identify cortical ROIs that encode different constituent representations within the hierarchical task (Fig 6A). In each cortical ROI, we fitted 5 representation similarity models (Fig 6C) and other control regressors (see Materials and methods for details) to assess different representations. The representation models encompass the following: Context (indicating whether the cue was classified as filled or

hollowed), Context × Shape, Context × Color (reflecting whether the task-relevant feature of the cue is shape-based or color-based depending on the context), Task (indicating whether the cue is associated with a face or scene response rule), and Decision (specifying the integrated task representation guiding subject responded with their index or middle finger).

We found that different regions encode different constituent representations of the task. Specifically, context representation was encoded in the left rostral prefrontal cortex (PFC), the left inferior frontal gyrus, the left dorsal medial PFC, the left temporoparietal junction, the left middle and the inferior temporal gyrus, the left precuneus, the right dorsal medial PFC, and the right middle temporal gyrus. Task-relevant feature representations given the task context (Context × Color and Context × Shape) were encoded in the left rostral PFC, the medial dorsal PFC, the sensorimotor cortex, the precuneus, the temporal cortex, and the occipital cortex. We did not observe positively significant cortical ROIs related to task performed representation. Decision representation was encoded in the dorsal lateral PFC (DLPFC), the insular cortex, the sensorimotor cortex, the intra parietal sulcus, and the anterior temporal cortex (Fig 6C).

Having determined the cortical regions that preferentially encode different representations, we then asked if thalamocortical interactions can predict voxel-wise activity patterns within these regions (Fig 6B), and whether there is a difference in prediction between different task switching conditions. To address this question, we performed voxel-to-voxel thalamocortical activity flow mapping analysis for cortical ROIs that encode different levels of task representations. We first compared our voxel-to-voxel thalamocortical interaction model to the 2 null models: the first assumed no spatial pattern of the thalamic evoked response carried task-related information, while the second assumed no thalamocortical FC pattern carried task-related information. We found that our thalamocortical interaction model for all 3 task-switching conditions outperformed the 2 null models across cortical regions representing different levels of task representations (see S6 Fig). The results suggest that the thalamocortical interaction model has better than chance performance on predicting different levels of cortical task representations during the hierarchical task-switching. However, we found no significant difference in prediction accuracy between EDS, IDS, and Stay conditions; therefore for the next prediction analysis, we used the evoked responses from all conditions combined, which models across these 3 conditions.

We then compared the prediction accuracy between ROIs that encode different levels of task representations. The goal is to determine whether the thalamocortical model can better predict cortical regions that encode a specific constituent representation. One-way rmANOVA showed that there is a significant difference between different cortical representations ($F$ (3,174) = 31.62, $p < 0.0001$, $\eta^2 = 0.16$; Context (mean ± SD): 0.36 ± 0.33, Context × Color (mean ± SD): 0.20 ± 0.34, Context × Shape (mean ± SD): 0.19 ± 0.35, Decision (mean ± SD): 0.26 ± 0.35; Fig 6D). Specifically, the thalamocortical model displayed highest prediction accuracy in predicting context representation (Context versus Context × Color: $t(58) = 6.04$, $p < 0.0001$, Cohen's d = 0.87; Context versus Context × Shape: $t(58) = 7.24$, $p < 0.0001$, Cohen's d = 1.04; Context versus Decision: $t(58) = 5.80$, $p < 0.0001$, Cohen's d = 0.51), then followed by predicting decision cortical representation (Decision versus Context × Color: $t$ (58) = 2.94, $p = 0.03$, Cohen's d = 0.41; Decision versus Context × Shape: $t(58) = 4.51$, $p = 0.0002$, Cohen's d = 0.60). We did not find significant difference of prediction accuracy between the 2 feature cortical representations (Context × Color versus Context × Shape: $t(58)$ = 2.49, $p = 0.09$, Cohen's d = 0.20).

To further test how different thalamic subregions influence cortical representations, we performed a virtual lesion analysis. We found that removing thalamic voxels from the thalamocortical activity flow model resulted in a decrease in the model's ability to predict voxel-wise activity patterns in cortical ROIs that encode all levels of hierarchical task representations

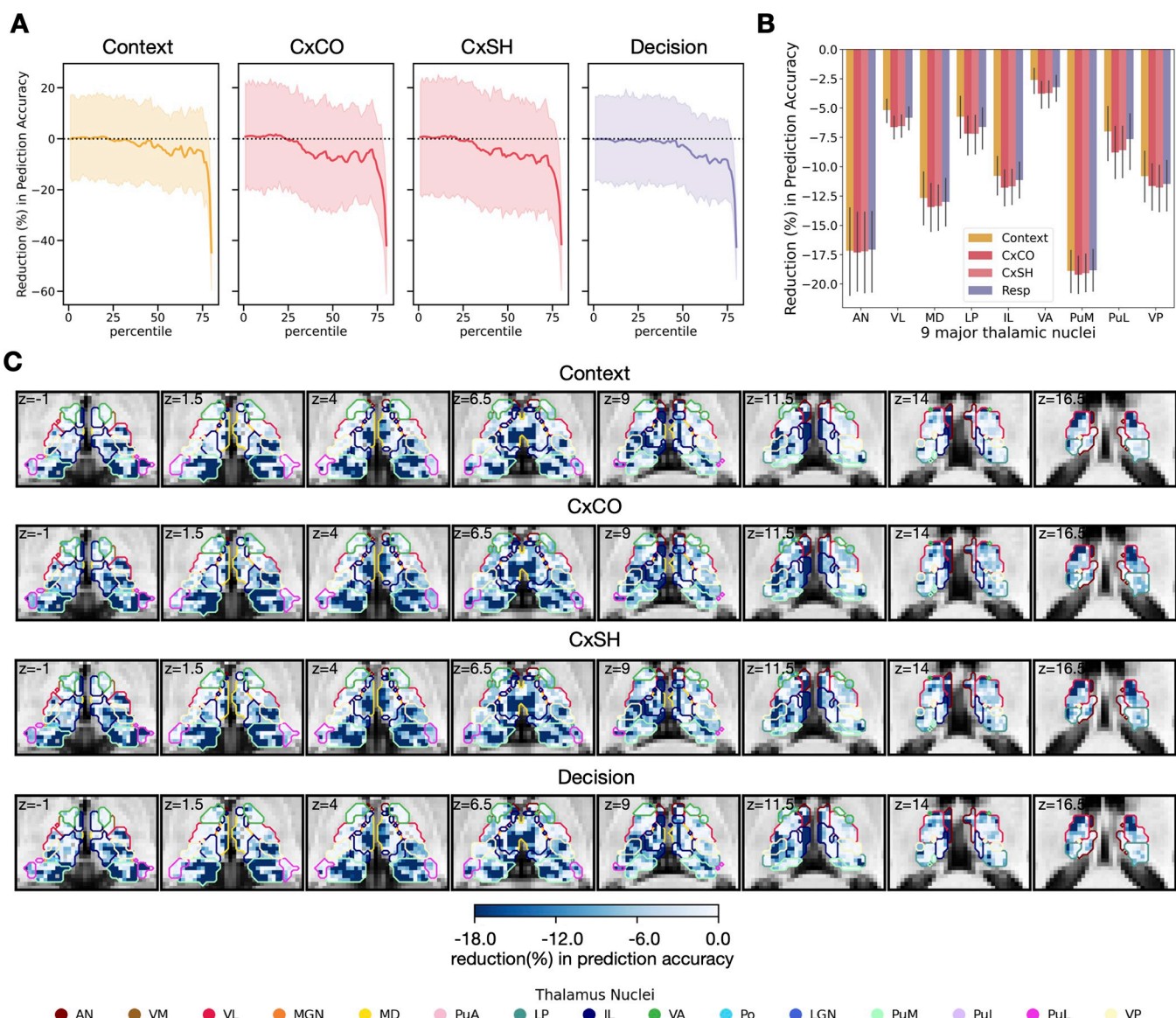

**Fig 7. Thalamic topography of reduction in model prediction.** (A) The group averaged results of thalamic topography of reduction in model performance when predicting various cortical representations. (B) Reduction in prediction accuracy for the 4 cortical representations. The error bar represents the 95% confidential interval. (C) Reduction in prediction accuracy for the different cortical representations after lesioning thalamic voxels ranking by the amplitudes of evoked responses, ranging from the 1st to the 80th percentile. Data used for (A) and (B) can be found in S1 Data, specifically in the sheets labeled "Fig 7A" and "Fig 7B." The group statistical maps presented in (C) can be accessed at https://identifiers.org/neurovault.collection:18728.

(Fig 7). We found that removing voxels from the anterior, the ventrolateral, the mediodorsal, the intralaminar, and the medial pulvinar, led to a consistent reduction in model's performance across different levels of cortical representations.

## Evoked activity in the basal ganglia (BG) and BG-cortical activity flow analysis

Our findings have demonstrated that thalamocortical interactions can predict activity patterns associated with hierarchical cognitive control. Another subcortical region likely critical for hierarchical control is the BG [3,15]. The BG has been hypothesized to inhibit or disinhibit

thalamocortical circuits when cortical representations need to be updated; however, it does not have direct projections to the cortex; therefore, the cortical representations of task information is likely to involve thalamocortical interactions that are potentially under the influence of the BG inhibitory output. It is unknown whether the thalamus and the BG will show similar or distinct response and activity flow prediction patterns; therefore, we investigated the role of the BG in predicting cortical activity patterns. Our prediction was that the BG will exhibit a similar response profile as the thalamus given the known frontal-BG involvement in hierarchical cognitive control [3,15]. However, considering that the output of BG, especially the globus pallidus nucleus, is mediated by the thalamus to influence cortical functions, we predicted that it would be less effective at predicting cortical activity patterns compared to the thalamus.

We first investigated how hierarchical task-switching modulates evoked responses in the BG. We found that the caudate nucleus and the putamen showed significant evoked responses to different hierarchical task-switching conditions (EDS, IDS, and Stay conditions; unthresholded: Fig 8A; thresholded activity maps at cluster corrected $p < 0.05$: Fig 8B) but not the globus pallidus. When comparing these conditions, the caudate nucleus displayed a stronger evoked response for EDS-IDS and EDS-Stay contrasts (Fig 8C). No significant clusters were observed for the IDS-Stay contrast. Similar to the patterns observed in the thalamus and the cortex, as the demand for hierarchical cognitive control increased, the caudate nucleus exhibited much stronger activity during EDS compared to IDS and Stay conditions. The overall BG response pattern remained consistent when considering trials where the current trial's response and cue were repeated from the previous trial (see S7 Fig).

While the BG also exhibited evoked response modulations by different hierarchical task-switching conditions, we expect the BG-cortex interaction model to have weaker performance in predicting task-related cortical activity when compared to the thalamocortical model. This is because the BG modulates thalamocortical interactions without directly projecting to the cortex. A BG-cortical activity flow model can therefore serve as a comparison model for our thalamocortical model. Instead of using the thalamic evoked response and thalamocortical FC matrix, we predicted the voxel-wise cortical activity pattern for each of representation cortical areas by calculating the dot product of voxel-wise BG evoked response and voxel-wise FC between BG voxel and every voxel within any given cortical region (BG model performance, see S8 and S9 Figs). The resulting predicted cortical activity pattern was compared with the observed cortical activity pattern using Pearson correlation, normalized by the noise ceiling (see Materials and methods for details). Consistent with our prediction, our findings demonstrated that thalamus model exhibited superior performance to all BG-cortical models when predicting cortical representations that encodes integrated task representation (Decision; Fig 8D). When predicting cortical representations encoding contextual (Context) and relevant feature (Context × Color, Context × Shape) representations (Fig 8D), thalamus outperformed putamen and globus pallidus model (Fig 8D).

## Discussion

The goal of the current study is to elucidate the role of the human thalamus in hierarchical cognitive control. Specifically, our study aimed to investigate the role of the thalamus in encoding and updating task representations within a multilevel hierarchical task structure, encompassing context, task-relevant features, and decision rules. We made 3 key findings. First, we observed increased evoked activity in the anterior, the medial, the mediodorsal, and the posterior thalamus during hierarchical cognitive control, particularly in response to updating contextual representations. Second, these thalamic subregions predominantly encoded context representations within the hierarchical task organization. Finally, we found that these thalamic

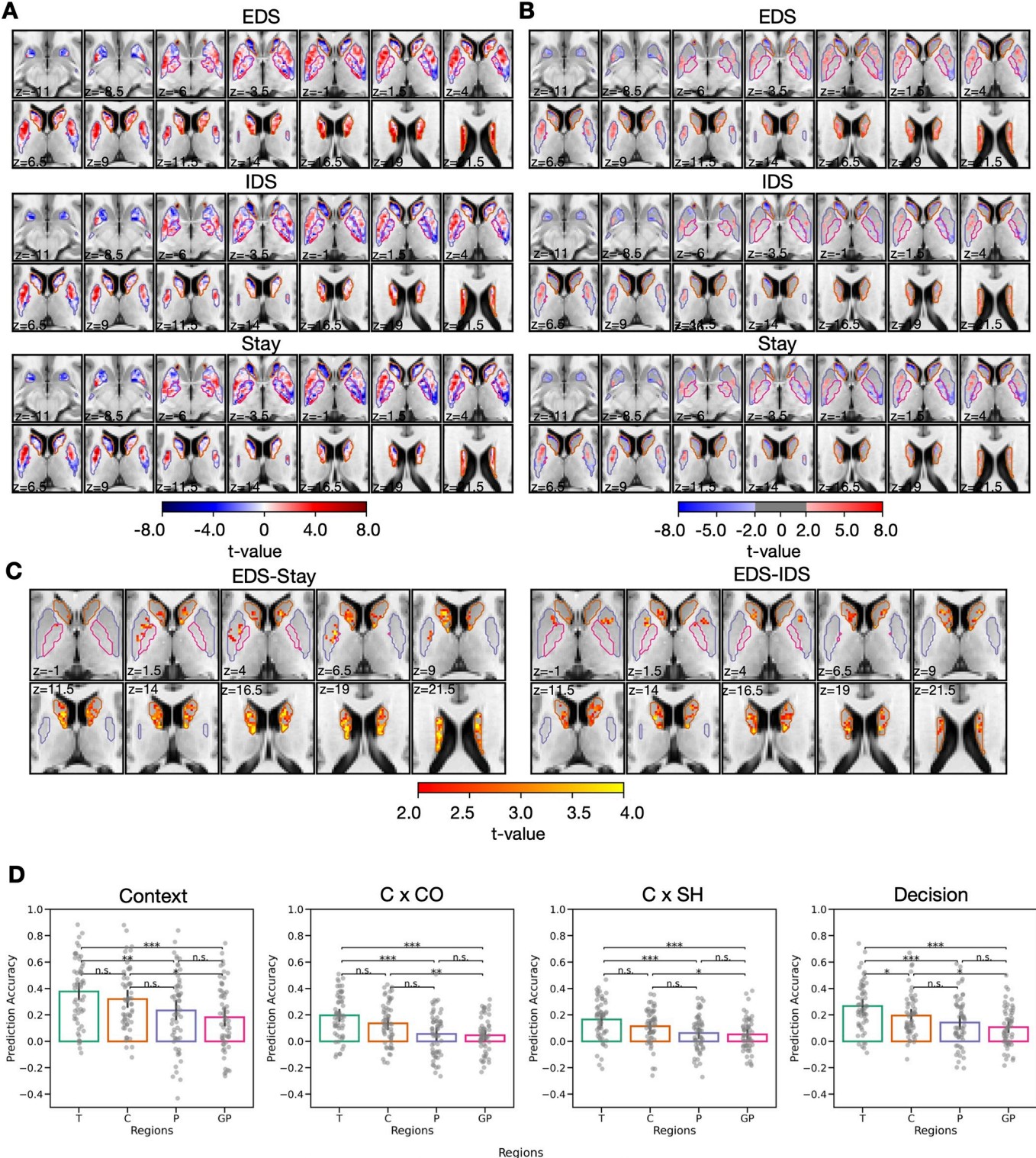

**Fig 8. BG involvement into hierarchical task switching.** (A) Unthresholded BG evoked responses (t-value) in 3 hierarchical task-switching conditions. (B) Thresholded BG evoked response in the 3 hierarchical task-switching conditions. (C) Statistically significant clusters in BG evoked responses were observed in EDS-Stay and EDS-IDS contrasts. There were no statistically significant results of IDS-Stay. (B, C) The results were first thresholded at a voxel-level threshold of $p < 0.05$, followed by cluster correction procedure with a cluster level threshold of $p < 0.05$, only showing clusters with a minimum cluster size (k) of 58 voxels. Clusters were defined as groups of voxels that are connected by sharing a face with their neighboring voxels. (D) Comparing the prediction accuracy of the 3

conditions between thalamocortical and BG-cortical models for parcellations identified from Fig 6C. The error bar represents the 95% confidential interval. Data used for (D) can be found in S1 Data, specifically in the sheet labeled "Fig 8D." The group statistical maps presented in (A–C) can be accessed at https://identifiers.org/neurovault.collection:18728.

regions interacted primarily with cortical areas that encode context representation, including the rostral PFC, the medial PFC, the temporoparietal junction, and the temporal cortex. Specifically, thalamocortical interactions were able to predict voxel-wise activity patterns in these cortical regions. Our results suggest that the thalamus, along with its associated thalamocortical interactions, encode and update context representations during hierarchical cognitive control.

The representational structure of hierarchical cognitive control enables adaptive and flexible cognitive control. Task representations consists of numerous constituent representations that are at least partially encoded by different cortical regions [2,40]. In this study, we designed a task where subjects performed with a hierarchically organized task representation. Critically, our design allowed us to dissociate different representations without comparing conditions with varying stimuli complexity, difficulty, or response requirements [5]. With this approach, we mapped cortical areas that encode different levels of task representations. We identified that the context-level representations were predominately encoded in the rostral PFC, the medial PFC, the temporoparietal junction, and the temporal cortex. Task-relevant feature representations were found to be encoded in the left rostral PFC, the medial dorsal PFC, the sensorimotor cortex, the precuneus, the temporal cortex, and the occipital cortex. Furthermore, representations pertaining to the decision of the task were localized in the DLPFC, the insular cortex, the sensorimotor cortex, the intra parietal sulcus, and the anterior temporal cortex. Given that cortical functions are intimately linked to the thalamus [29], our study investigates the function and representation encoded by the thalamus within this characterized hierarchical task structure.

Results from our MVPA analysis showed that the thalamus encodes contextual information. However, we did not find evidence that other task-relevant information within the hierarchical task structure is encoded in the thalamus. This may be due to the limited resolution or statistical power of functional magnetic resonance imaging (fMRI) data to detect such signals. Notably, several thalamic nuclei, including the anterior, the ventroanterior, the ventrolateral, the mediodorsal, the intralaminar, the medial pulvinar, and the lateral posterior nuclei, showed strong context coding. This findings is consistent with prior studies of human imaging and non-human animal models, which found that in behavioral tasks where subjects have to apply distinct response strategies under different contexts, the mediodorsal thalamus primarily encode context representations [26,27].

The selective encoding of context representation by the thalamus likely constrains how thalamocortical processes support hierarchical control. We found that the several thalamic nuclei also exhibited stronger evoked responses during task-switching conditions that involved updating the context representation (EDS). Significant task-evoked activity in multiple thalamic nuclei, including the anterior, the ventrolateral, the mediodorsal, the lateral posterior, the intralaminar, the ventroanterior, the medial pulvinar, the lateral pulvinar, and the ventral posterior nuclei. These thalamic nuclei showed most pronounced increase in evoked activity when subjects updated the highest-order task context. Furthermore, task-switching that updated other constituent representations within the hierarchical task structure did not elicit stronger evoked responses in the thalamus, further demonstrating the specificity of thalamic contextual coding.

A recently developed computational model has provided a hypothetical mechanism for how the thalamus can encode contextual representations [26]. In this model, neurons in the

mediodorsal thalamus receive converging inputs from prefrontal neural populations that encode different high-dimensional representations of task rules. Through this converging input, thalamic neurons extract a summary signal from multiple converging cortical inputs, encoding a low-dimensional, compressed representation abstracted from the high-dimensional inputs. This observation is further consistent with our prior fMRI studies, where we found that relative to the cortex, the thalamus encodes low-dimensional structures of cognitive activity [32], and that thalamic subregions showing the strongest context coding also exhibit converging connectivity (hub-like connectivity) with multiple frontoparietal structures [30,33].

In the cortex, we observed stronger evoked responses in several frontoparietal regions during hierarchical task switching. Our findings are consistent with prior research [2,15,40,41], suggesting that updating higher-level contextual information engages regions such as the rostral PFC, the posterior parietal, the posterior cingulate, and the insular areas, while updating lower-level task-relevant information predominantly involves the premotor and the parietal regions. To further investigate how thalamic functions may interact with these cortical regions, we developed a novel activity flow mapping analysis [34–36]. This method models thalamic evoked responses and thalamocortical connectivity patterns as input features to predict cortical activity patterns. We then compared our model with 2 null models, which posited that neither the thalamus nor thalamocortical connectivity carries task-specific information. We found that our thalamocortical interaction model significantly outperformed the 2 null models, successful in predicting cortex-wide activity patterns associated with our task.

To further determine the specific cortical task representation influenced by thalamocortical interactions, we adapted the model to predict voxel-wise activity patterns within cortical ROIs encoding different levels of task representation. Our results showed that the thalamocortical model most accurately predict voxel-wise activity patterns in cortical ROIs that encode the context representation, outperforming ROIs encoding other representations within the hierarchical task structure. This finding is further consistent with our prior findings demonstrating the thalamic preference for encoding the context of hierarchical task representations and showed increased activity during task-switching conditions that involved context update.

A logical question concerns which thalamic subregions contributed to predicting context-related cortical activity patterns? Subsequent simulated lesion analysis revealed that certain thalamic subregions had stronger contribution, particularly the anterior, the ventrolateral, the mediodorsal, the intralaminar, and the medial pulvinar nuclei. The removal of these thalamic nuclei from the thalamocortical model led to a decrease in the model performance, and the decrement was more pronounced than other thalamic subregions. These thalamic subregions contributing most to activity flow model performance spatially align with those whose evoked responses were mostly strongly modulated by contextual task switching. We speculate that these identified thalamic subregions are particularly well-suited for contextually guided cognitive control due to their hub-like connectivity property, capable of extracting context representations via its converging inputs, and ideally positioned to influence cortical representations via its distributed connectivity [30]. Indeed, we found that removing these thalamic subregions from the activity flow prediction model also negatively affected the prediction of activity patterns in ROIs encoding other task-related features. This finding suggests that in cortical regions not strongly encoding context representations, receiving contextual influence from the thalamus may also be critical for the selective processing of task-relevant features. Anatomically, it is long known that the anterior, the mediodorsal, and the pulvinar nuclei have a strong anatomical relationship with frontoparietal regions that we found to encode context representations [23–25]. For example, the anterior and mediodorsal thalamus have projections to both the lateral and medial prefrontal cortex, including the medial superior frontal gyrus, the

superior frontal gyrus, and the middle frontal gyrus [42–44]. The pulvinar nucleus mainly projects to the prefrontal, the posterior parietal, the temporal, and the extrastriate areas [25,45–47].

The BG, also a crucial part of the cortico-striatal-thalamic circuit, is proposed to act as a gate for updating task representation in the cortex [12,14,15,48]. While the basal ganglia's response to hierarchical task-switching was found to be similar to that of the thalamus, its ability to predict cortical activity patterns was significantly weaker than the thalamocortical model. Our previous work also observed a better performance for the thalamocortical model when compared to the BG-cortical model in predicting the cortical activity related to a working memory task [31] and cognitive tasks across multiple behavioral domains [32]. These results suggested that the thalamus has a more direct impact on cortical activity pattern than BG in predicting the hierarchical task representations. In the traditional BG-thalamic gating model, the ventrolateral and ventroanterior nucleus is the major target of BG inhibitory output [22,49]. Other thalamic nuclei, including several that we identified, such as the mediodorsal, the anterior, and the pulvinar nucleus, may also play an essential role in hierarchical task switching by receiving inhibitory outputs from other basal ganglia structures such as the substantia nigra [50] and the ventral pallidum [51,52]. More recently, in rodent models, it has also been demonstrated that the mediodorsal thalamus is capable of directly inhibiting and exciting cortical context representations [27]. Incorporating these findings with our own, the thalamocortical interaction likely plays an active role beyond merely receiving inhibitory output from BG. While we found that thalamocortical interactions involving multiple thalamic subregions can predict cortical activity patterns associated with context representations, due to the limitations of fMRI, we were not able to discern whether it is exciting or inhibiting cortical representations under different task conditions.

To summarize, we provided a detailed characterization of how the human thalamocortical system contributes to hierarchical cognitive control. Our results are consistent with the interpretation of the thalamus as a central and active node contributing to brain-wide information processes. The functional hubs within the thalamus, which specialize in processing contextual information, play a key role in encoding and updating context representations to facilitate flexible, contextually guided cognitive control. This function is vital for the brain's capacity to adapt and respond to evolving task requirements.

## Materials and methods

### Participants

Seventy-four healthy adults participated in our fMRI study. Fifteen participants were excluded from further data analysis based on the following criteria: 5 participants were excluded because of poor behavioral performance (accuracy <0.7); 1 participant was excluded because of failure in the response equipment; 1 participant voluntarily dropped out midway through the experiment; 8 participants were rejected for excessive head motion (mean framewise displacement (FD) mean > 0.4 and more than 33.3% of data removed because of excessive head motion). Therefore, a total of 59 participants (44 females and 15 males, age: 22.32 ± 4.71) were included for data analyses. All participants were right-handed, had normal or corrected-to-normal vision, and reported no history of a neurological conditions.

### Ethics statement

The study protocol was reviewed and approved by Institutional Review Boards at the University of Iowa (IRB-01-201808855). All participants provided written informed consent in

accordance with the approved procedures. The study was conducted in compliance with the ethical principles expressed in the Declaration of Helsinki.

## Hierarchical cognitive control task

In our behavioral task, participant utilized a hierarchically structured task representation to determine which response rule to adopt to respond to the probe stimuli [5]. In each trial, one of 8 different colored geometric cues was presented for 0.5 s and then was immediately followed by a probe image of either a face or a scene. The probe image was presented for 2.5 s. Participants were asked to respond during this time window. Trials were separated by a jittered inter-trial-interval that varied between 1.5 s to 10.5 s (average: 4.5 s) with a "+" at the center of monitor (Fig 1A). Participants were required to respond to the probe picture based on one of 2 response rules: judging whether the picture depicted a face (face rule) or a scene (scene rule). The correct response rule depended on 3 attributes of the cue (fill texture: solid or hollow, shape: square or circle, color: blue or red), and these attributes were organized in a hierarchical fashion (Fig 1B). The highest level of attributes, referred to as the "context," evaluated the fill texture of the cue (solid versus hollow). This context information determined which mid-level "feature" participants should utilize to determine the response rule. Specifically, a solid cue prompted the participant to judge whether the shape of the cue was a circle or a square, while a hollow cue instructed them to judge whether the color of the cue was blue or red. This mapping was counterbalanced across participants. Each feature from the mid-level was then associated with either the face or scene rule. For example, a circle corresponded to the face rule, while a square corresponded to the scene rule. Once the correct response rule was determined, participants pressed a button to indicate "yes" or "no" based on the probe picture. The responding buttons were counterbalanced across participants. The sequence of cues was fully randomized across trials, in which all 8 cues were randomly and equally distributed across trials. Participant completed 8 runs (about 9 min/run) in total, and each run contained 51 trials.

The hierarchical structure of our behavioral task allowed us to introduce 3 levels of task-switching operations that update task representations: EDS, IDS, and Stay (Fig 1A and 1B). For EDS, participants were required to update the highest-level of task context. IDS, on the other hand, switched between task-relevant features under the same task context. Lastly, during Stay trials participants maintained the same response rule for 2 consecutive trials. Each participant performed 204 Stay trials, 102 EDS trials, and 102 IDS trials.

## Magnetic resonance imaging

All participants were scanned at the Magnetic Resonance Research Facility at the University of Iowa, utilizing a GE SIGNA Premier 3T MRI Scanner with a 48-channel head coil. Structural images were acquired using a multi-echo MPRAGE sequence (TR = 2,348.52 ms; TE = 2.968 ms; flip angle = 8˚; field of view = 256 * 256; 200 sagittal slices; voxel size = 1 mm$^3$). Functional images were acquired using an echo-planar sequence sensitive to blood oxygenated level-dependent (BOLD) contrast (TR = 1,800 ms; TE = 30 ms; flip angle = 75˚; voxel size = 2.5 mm$^3$).

## fMRI preprocessing

All fMRI data were preprocessed using fMRIPrep version 20.1.1 [53] to reduce noise and transform the data from the subject's native space to the ICBM 152 Nonlinear Asymmetrical template version 2009c for group analysis [54]. The preprocessing steps included bias field correction, skull-stripping, co-registration between functional and structural images, tissue segmentation, motion correction, and spatial normalization to the standard space. As part of the preprocessing, we included rigid-body motion estimates, cerebral spinal fluid (CSF), and

white matter (WM) noise components obtained through the component-based noise correction procedure [55] in the regression model to reduce the influences from noise and artifacts. We did not perform any spatial smoothing.

### Behavioral data analysis

We performed one-way repeated measure ANOVA with 3 levels (EDS, IDS, and Stay) to examine the effects of hierarchical task-switching on response time and accuracy. Incorrect trials were not included in for the reaction time analysis. We applied Bonferroni's correction method with a family-wise error rate (FWER) of 0.05. We reported Cohen's d as the metric of effect size.

### fMRI data analysis

**Evoked responses.** The analysis of evoked responses was conducted using AFNI [56]. To estimate the evoked response for individual subject, we performed voxel-wise general linear model (GLM) analysis, which combined a generalized least squares regression with voxel-wise restricted maximum likelihood (REML) estimate of an ARMA (1,1) temporal correlation model (3dDeconvolve and 3dREMLfit). For each voxel, the GLM was constructed and fitted to the preprocessed time series with the task regressors and nuisance regressors (see fMRI preprocessing section). All task regressors were modeled with 9 basic TENT functions that lasted from time 0 to 14.4 seconds post cue, separately for each condition. High motion volumes (FD > 0.4 mm) were removed via the censoring option in 3dDeconvolve, and subjects with more than 33.3% of the data censored were dropped from further analysis. The group analysis involved examining the 3 task-switching conditions (EDS, IDS, and Stay), as well as contrasting pairs of task-switching conditions (EDS-IDS, IDS-Stay, and EDS-Stay). This was performed using a linear mixed model with a local estimate of random effects variance (3dMEMA). To correct for multiple comparisons, we used a whole-brain, family-wise error (FWE) cluster correction procedure (3dClustSim), where the smoothness was estimated using the updated spatial autocorrelation function option to reduce false positives [57]. Clusters were formed by first applying a per-voxel $p$-value threshold of 0.05. This statistical significance of each cluster was determined using a cluster-level threshold of $p < 0.05$, with a minimum cluster size (k) of 58 voxels to ensure that clusters reflect true positive signals while controlling for FWE across the whole brain. The visualization of cortical result was realized using Nilearn's surface.vol_to_surf function, which projects the volumetric data onto the surface for display purpose.

**Functional connectivity (FC).** To estimate task-state thalamocortical FC, we calculated the background connectivity [39] between thalamus and cortices by using the residual from the GLM procedure (see Evoked responses section). For each participant, we extracted voxel-wise signals (using residuals from GLM) from the thalamus, based on the Morel Atlas [37], and from 400 cortical regions of interest [38] and then performed principal component regression analysis. Subsequently, a principal component analysis (PCA) was applied to the fMRI time series across all thalamic voxels, using the maximum available number of components (for example, 199 components when 200 time points were present). These components were entered as predictors and cortical ROI data as the dependent variable in a linear regression model. The resulting beta coefficients were then transformed back to original thalamocortical space, which described the linear relationship between thalamus time series and cortical time series.

### Multivoxel pattern analysis (MVPA)

We aimed to understand the level at which the thalamus preferentially classifies task-relevant information in a hierarchical structure. We performed an MVPA approach, using a spherical

searchlight procedure with a radius of 6 mm$^3$. We first removed the time series of noise by applying nuisance regressors (see Preprocessing section) for data modeling (3dTproject in AFNI). The Least Squares Sum method (3dLSS in AFNI) was then used to model the trial-by-trial evoked responses for each of the 8 cues. For each searchlight region, we performed a logistic regression classifier to decode the multivoxel activity pattern associated with the context (solid versus hollow), color (blue versus red), shape (square versus circle), and task (face versus scene), respectively. Our cross-validation strategy involved a leave-one-group-out approach, such that one functional run of data was reserved as test data while the classifiers were trained on other runs. To assess our findings at the group level, we compared the group average decoded results against an empirically determined null-distribution, using a cluster correction procedure to correct for multiple comparisons [58]. The individual voxels were thresholded at $p < 0.001$, and only clusters that survived a cluster correction threshold of $p < 0.001$ were reported. The minimum cluster size was 343 voxels.

## Activity flow mapping analysis

**Activity flow mapping analysis framework.** We modified the activity flow mapping procedure [34–36] to determine whether thalamocortical (or BG-cortical) interactions can predict cortical task activity patterns during hierarchical cognitive control. This activity flow model was calculated by computing the dot product between the seed region (e.g., thalamus or BG) evoked response vector and the FC matrix between the seed region and cortices (e.g., thalamocortical FC or BG-cortical FC):

$$P_{ctx} = O_{seed} \cdot WFC_{SeedCtx}$$

Here, $P_{ctx}$ refers to the vector of predicted parcellation or voxel-wise cortical evoked response for the voxel-to-parcellation and voxel-to-voxel activity flow procedure, respectively. $O_{seed}$ corresponds to the vector of the voxel-wise seed region (e.g., thalamus or BG nuclei) evoked responses, which is the task-specific beta value estimated using subject-level voxel-wise GLM estimates. $WFC_{SeedCtx}$ refers to FC matrix between the seed region and the cortices (e.g., thalamocortical FC or BG nuclei-cortical FC), which is the beta value estimated using subject-level principal component regression analysis.

The model performance was evaluated by the prediction accuracy from each of activity flow mapping procedures. Prediction accuracy for the 3 hierarchical task-switching conditions (EDS, IDS, and Stay) was evaluated using split-half cross validation. We calculated Pearson's correlation between z-transformed observed and predicted cortical evoked responses. Data were split into 2 parts and correlations were computed between observed evoked responses from the first half and predicted evoked responses from the second half, then vice versa. The average of these 2 correlations gave us the final prediction accuracy.

**Noise ceiling.** Note that since the upper bound for the prediction accuracy is limited by the measurement noise of cortical and thalamic evoked responses, prediction accuracy was further normalized by the split-half noise ceiling [59], taking into account the split-half model reliability:

$$r_{nc} = \frac{\sqrt{vU}}{\sqrt{vU + vE}}$$

Here, $r_{nc}$ refers to the split-half noise ceiling metric, and $vU$ is the variance of true signal, the predicted cortical patterns in first 4 runs, whereas $vE$ is the variance of noise, the difference of predicted cortical patterns between first and last 4 runs. The reported split-half prediction accuracy was calculated as the split-half cross-validated prediction accuracy normalized by

split half noise ceiling. We did not observe any significant difference of noise ceiling metric across 3 hierarchical task-switching conditions for any type of activity flow procedure (see S2 Text).

**Null models.** To evaluate the performance of the activity flow mapping analysis, we compared the prediction accuracy of the model to 2 null models. The first null model was constructed by randomly shuffling each subject's seed region evoked responses ($O_{seed}$ in the equation) with 5,000 random permutations. The second null model was constructed by randomly shuffling elements within the FC matrices between the seed region and cortices ($WFC_{SeedCtx}$ in the equation) with 5,000 random permutations. These 2 null distributions satisfied the null hypothesis that neither the spatial pattern of seed region evoked responses nor the patterns of FC between seed region and cortices carry task-specific information.

**Effect of simulated thalamic lesions on activity flow mapping.** To determine which thalamic subregions contribute to predicting cortical activity patterns, we systematically removed 20% of thalamic voxels from the activity flow mapping analysis based on the strength of evoked response amplitudes (the absolute value ranked from weakest to strongest). We stepped the removal from the 1st to 80th percentile in 1 percentage increments. Activity for removed voxels was set to zero before calculating the predicted patterns. The reduction percentage in prediction accuracy was calculated as the difference of prediction accuracy before and after the removal of thalamic voxels.

## Thalamocortical interactions and cortical control representations

**Representational similarity analysis (RSA).** To investigate brain regions involved in encoding different constituent representations within the hierarchically structured task representations, we performed ROI-based RSA looping through 400 cortical ROIs [38]. We initially conducted voxel-wise GLM analysis, using 9 basic TENT functions to model the impulse response of 8 different cues from time 0 to 14.4 seconds following each cue, as implemented using the 3dDeconvolve. Within the ROI, we averaged the signals over 9 TRs for each voxel and then calculated the neural dissimilarity between voxel-wise patterns for each pair of cues (8 cues in total). The neural dissimilarity was computed as the spatial Mahalanobis distance. We then fitted the following representational models onto the observed dissimilarity matrix: (1) Context: whether the cue is classified as filled or hollowed. (2) Task-relevant feature: the interaction between context and color; the interaction between context and shape. (3) Task performed: whether the cue is associated with the face or scene response rule. (4) Decision: specifying the integrated task representation guiding subject responded with their index or middle finger. (5) Other control regressors include the main effect of shape and color; the two-way interactions: shape × task performed; color × task performed; color × shape; context × task performed; the three-way interactions: context × shape × task performed; context × color × task performed.

The lower triangle of the neural dissimilarity matrix was extracted and regressed onto an intercept and the lower triangle of the representational models, resulting in one beta coefficient per variable, per subject, per ROI. Group-level analyses were conducted using $t$ tests across subjects, with the null hypothesis that the beta value equals zero. To control for multiple comparisons, we applied false discovery rate (FDR) correction (q < 0.05).

**Voxel-to-voxel thalamocortical activity flow on cortical regions representing hierarchical task representation.** To test whether the thalamocortical activity flow model can predict cortical regions that found to encode specific representations, we conducted RSA to identify cortical regions responsible for encoding context, task-relevant features, task performed, and decision. Subsequently, we tested whether the thalamocortical activity flow model could successfully predict the voxel-wise activity patterns in the identified ROI for the EDS, IDS, Stay,

and All (which models evoked response of all task-switching conditions) conditions. We first compared the model performance on these 4 conditions relating to hierarchical switching (EDS, IDS, Stay, and All conditions) to 2 null models, respectively (see Null models). We then performed a one-way repeated measure ANOVA to examine the effects of hierarchical task switching (EDS, IDS, and Stay conditions) on model performance. To examine differences in model performance across different cortical representations, we performed a one-way repeated measure ANOVA specifically on the All condition. We applied Bonferroni's correction method with an FWER of 0.05. We reported Cohen's d as the metric of effect size. To determine which thalamic subregions contribute to predicting cortical activity patterns of different levels of task representation, we performed simulated thalamic lesions on activity flow mapping (see Effect of simulated thalamic lesions on activity flow mapping section).

## Supporting information

**S1 Fig. Behavioral results considering response repetition and cue repetition.** (A) Behavioral results for response switching. (B) Behavioral results for response repetition. (C) Behavioral results for cue repetition. (A–C) The left panel is reaction time and the right panel is accuracy. *** $p < 0.001$; ** $p < 0.01$; n.s., nonsignificant. The error bar represents the 95% confidential interval. The black dot indicates the mean value, while the colored dot represents data from individual subjects. Lines connect the data points for each subject across different conditions. Data used for (A–C) can be found in S1 Data, specifically in the sheet labeled "S1A Fig," "S1B Fig," and "S1C Fig."
(TIFF)

**S2 Fig. Cortical evoked response to hierarchical task-switching considering response repetition and cue repetition.** (A) Contrasts between hierarchical task-switching conditions, accounting for trials where the current trial's decision (choosing "yes" or "no") repeats from the previous trial (RR). (B) Contrasts between task-switching conditions, separate trials where the cue of the current trial either repeats (CR) or switches (CS) from the previous trial. The results were first thresholded at a voxel-level threshold of $p < 0.05$, followed by cluster correction procedure with a cluster level threshold of $p < 0.05$, only showing clusters with a minimum cluster size (k) of 58 voxels. Clusters were defined as groups of voxels that are connected by sharing a face with their neighboring voxels. The group statistical maps presented in S2 Fig can be accessed at https://identifiers.org/neurovault.collection:18728.
(TIFF)

**S3 Fig. Hemodynamic response functions (HRFs) of major thalamic nuclei.** Data used for this figure can be found in S1 Data, specifically in the sheet labeled "S3 Fig."
(TIFF)

**S4 Fig. Thalamic voxel-wise evoked responses to hierarchical task-switching considering response repetition and cue repetition.** (A) Contrasts between conditions, trials where current trial's decision (choosing "yes" or "no") repeats the previous trial (RR). (B) Contrasts between task-switching conditions, separate trials where the cue of the current trial either repeats (CR) or switches (CS) from the previous trial. The results were first thresholded at a voxel-level threshold of $p < 0.05$, followed by cluster correction procedure with a cluster level threshold of $p < 0.05$, only showing clusters with a minimum cluster size (k) of 58 voxels. Clusters were defined as groups of voxels that are connected by sharing a face with their neighboring voxels. The group statistical maps presented in S4 Fig can be accessed at https://identifiers.org/neurovault.collection:18728.
(TIFF)

**S5 Fig. Effect of spatial smoothing on thalamic result.** (A) Thalamic data without spatial smoothing in EDS-IDS condition, analyzed with a peak threshold of $p < 0.05$, followed by cluster corrected at $p < 0.05$ with a minimum cluster size of 58 voxels. (B) Thalamic data smoothed with an 8 mm kernel in EDS-IDS condition, using a peak threshold of $p < 0.001$, followed by cluster correction at $p < 0.05$ with a minimum cluster size of 42 voxels. Clusters were defined by whether adjacent voxels touch either in-plane or at points. The group statistical maps presented in S5 Fig can be accessed at https://identifiers.org/neurovault.collection:18728.
(TIFF)

**S6 Fig. Voxel-to-voxel thalamocortical interactional model.** (A) Thalamocortical interaction model compared to null models. (B) Model performance of 3 hierarchical task switching conditions when predicting different cortical representations. Data used for (A) and (B) can be found in S1 Data, specifically in the sheet labeled "S6A Fig" and "S6B Fig."
(TIFF)

**S7 Fig. BG voxel-wise evoked response to hierarchical task-switching, accounting for both response repetition and cue repetition.** (A) Contrasts between conditions, separating trials where current trial's decision (choosing "yes" or "no") repeats from the previous trial (RR). (B) Contrasts between task-switching conditions, where the cue in the current trial either repeats (CR) or switches (CS) from the previous trial. The results were first thresholded at a voxel-level threshold of $p < 0.05$, followed by cluster correction procedure with a cluster level threshold of $p < 0.05$, only showing clusters with a minimum cluster size (k) of 58 voxels. Clusters were defined as groups of voxels that are connected by sharing a face with their neighboring voxels. The group statistical maps presented in S7 Fig can be accessed at https://identifiers.org/neurovault.collection:18728.
(TIFF)

**S8 Fig. Voxel-to-voxel BG-cortical interaction model compared to 2 null models.** (A) Caudate-cortical interaction model compared to 2 null models. (B) Putamen-cortical interaction model compared to 2 null models. (C) Globus pallidus-cortical interaction model compared to 2 null models. Data used for this figure can be found in S1 Data, specifically in the sheet labeled "S8 Fig."
(TIFF)

**S9 Fig. Voxel-to-voxel BG nuclei-cortical interaction model performances for predicting hierarchical task-switching conditions and cortical representations.** (A) Model performance of 3 hierarchical task switching conditions when predicting different cortical representations. (B) Model performance of predicting different cortical representations. $^*p < 0.05$; $^{**}p < 0.01$; $^{**}p < 0.001$; n.s., not significant. Data used for this figure can be found in S1 Data, specifically in the sheet labeled "S9 Fig."
(TIFF)

**S1 Text. Brain regions activated during hierarchical task-switching.**
(DOCX)

**S2 Text. Noise ceiling metric.**
(DOCX)

**S1 Data. Source data of Figs 1C, 1D, 5A, 5D, 5E, 6D, 7A, 7B, 8D, S1, S3, S6, S8, and S9 and S2 Text.**
(XLSX)

## Acknowledgments

This work was conducted on an MRI instrument funded by 1S10OD025025-01.

## Author Contributions

**Conceptualization:** Xitong Chen, Kai Hwang.

**Data curation:** Xitong Chen, Stephanie C. Leach.

**Formal analysis:** Xitong Chen.

**Funding acquisition:** Kai Hwang.

**Investigation:** Xitong Chen.

**Methodology:** Xitong Chen, Kai Hwang.

**Project administration:** Juniper Hollis, Dillan Cellier.

**Resources:** Kai Hwang.

**Software:** Xitong Chen.

**Supervision:** Kai Hwang.

**Visualization:** Xitong Chen.

**Writing – original draft:** Xitong Chen.

**Writing – review & editing:** Xitong Chen, Kai Hwang.

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
