## [Editor Report · Decision Letter 0]

2 Jul 2024

Dear Dr Chen, 

Thank you for submitting your manuscript entitled "Thalamocortical contributions to hierarchical cognitive control" for consideration as a Research Article by PLOS Biology.

Your manuscript has now been evaluated by the PLOS Biology editorial staff as well as by an academic editor with relevant expertise and I am writing to let you know that we would like to send your submission out for external peer review.

Once your full submission is complete, your paper will undergo a series of checks in preparation for peer review. After your manuscript has passed the checks it will be sent out for review. To provide the metadata for your submission, please Login to Editorial Manager (https://www.editorialmanager.com/pbiology) within two working days, i.e. by Jul 04 2024 11:59PM.

Kind regards,

Christian

Christian Schnell, PhD

Senior Editor

PLOS Biology

cschnell@plos.org

---

## [Decision Letter · Decision Letter 1]

27 Aug 2024

Dear Dr Chen,

Thank you for your patience while your manuscript "Thalamocortical contributions to hierarchical cognitive control" was peer-reviewed at PLOS Biology. It has now been evaluated by the PLOS Biology editors, an Academic Editor with relevant expertise, and by several independent reviewers. 

In light of the reviews, which you will find at the end of this email, we would like to invite you to revise the work to thoroughly address the reviewers' reports.

As you will see below, the reviewers have overall very positive comments about your study. Reviewer 1 and 3 have a few questions about some of the analyses, while Reviewer 2 wants more clarity and information on some methodological and statistical aspects. He also recommends to improve some of the analyses.

Given the extent of revision needed, we cannot make a decision about publication until we have seen the revised manuscript and your response to the reviewers' comments. Your revised manuscript is likely to be sent for further evaluation by all or a subset of the reviewers.

**IMPORTANT - SUBMITTING YOUR REVISION**

*Re-submission Checklist*

*Published Peer Review*

*PLOS Data Policy*

*Blot and Gel Data Policy*

Sincerely,

Christian

Christian Schnell, PhD, 

Senior Editor

PLOS Biology

cschnell@plos.org

REVIEWS:

Reviewer #1 (Burkhard Pleger): This fMRI study explores the role of thalamic representation and thalamocortical interactions in hierarchical cognitive control in humans. The findings reveal that specific thalamic nuclei, including the anterior, mediodorsal, ventrolateral, and pulvinar, exhibit stronger responses during task context switching. Decoding analysis indicates that thalamic activity preferentially encodes task contexts within hierarchical task structures. To further understand the contribution of thalamocortical interactions to these representations, the researchers developed a thalamocortical functional interaction model. This data-driven model outperformed other models, particularly in predicting cortical activity patterns associated with context representations. 

These results highlight the crucial role of thalamic activity and thalamocortical interactions in facilitating contextually guided hierarchical cognitive control. The task is exceptionally well-designed to address the scientific question effectively. The study is novel and innovative, providing fresh insights into the underexplored role of the thalamus in cognitive processing. It significantly contributes to our understanding of how the thalamus supports complex cognitive functions through intricate brain network interactions.

I have only a few points I would like the authors to consider:

1. For MVPA, the authors conducted voxel-wise t-tests comparing the decoded results against a chance level of 0.5. In the results section, they report the highest decoding accuracy across almost all thalamic nuclei, including the anterior, ventroanterior, ventrolateral, mediodorsal, intralaminar, medial pulvinar, and lateral pulvinar. However, the mean accuracy for each nucleus was 0.52 ± 0.004. Despite significant t-tests this seems rather unconvincing to me.

2. Why authors haven't applied the RSA approach to dissect the thalamic nuclei according to the different representations of the hierarchical task structures. The methods described for identifying cortical representations through ROI RSA to fit five representation similarity models and assess different representations are thorough and insightful. It would be beneficial to understand if the different thalamic nuclei also encode these hierarchical task representations similarly.

3. As the authors correctly pointed out, the basal ganglia (BG) are hypothesized to regulate thalamocortical circuits by inhibiting or disinhibiting them when cortical representations require updating. However, since the BG lack direct projections to the cortex, it is likely that the cortical representations of task information involve thalamocortical interactions modulated by the inhibitory output from the BG. Given this framework, authors could directly test their hypothesis. Network models, such as non-linear Dynamic Causal Modeling (https://www.ncbi.nlm.nih.gov/pmc/articles/PMC2636907/), offer the possibility to directly test the inhibitory/excitatory influence of BG time series on thalamocortical connections.

Reviewer #2: Chen et al. describe an fMRI study of the contributions of the thalamus to hierarchical control. Some prominent models of these control mechanisms describe a PFC-basal ganglia-thalamic circuit that gates the flow of information to selectively activate representations based on hierarchically organized contingencies. As the authors point out, the role of the thalamus in this process has not been well-examined, with most models assuming its function to be equivalent to the parallel circuitry involved in controlling motor responses. Chen et al. demonstrate that parts of the thalamus respond to higher-order control demands and maintain a representation of the higher order contextual variable in a hierarchical control task. They also use 'activity flow mapping' to show that thalamic activity is predictive of cortical responses - mediated by functional connectivity between thalamic and cortical voxels. This study makes an important contribution to our understanding of control processes by providing a detailed examination of the thalamic circuitry that supports hierarchical control. The manuscript is well-written, clearly organized and timely. However, some important aspects of the methods and statistical choices were missing or not clear. I also have some suggestions and questions about analyses that I will describe below.

1. The authors convincingly show that task representations in the thalamus and cortex are related, but do not show any link between these representations and behavior. This might be somewhat out of the scope of this study, but it would be interesting to know if the thalamic context representation is correlated with task performance within-subject or across individuals?

2. In several figures (e.g. Fig 2, Fig 3c), the legend says that the image shows cluster corrected data at p < 0.05. However, the color bar indicates that the range is from t > 2.0, which is the t-value for p < 0.05, two-tailed for N = 59. This is an unusually low threshold for displaying fMRI results. It is more common to show data either at the cluster-corrected threshold, or at a threshold of p < 0.01 or p < 0.001, uncorrected for "visualization purposes." In any case, the authors need to make it clearer which of these effects survive cluster correction, and also report the k-value used to threshold any surviving clusters. The manuscript is also missing tables summarizing statistical effects that survive cluster correction and their coordinates in MNI space, which would be very useful for other researchers wishing to follow up on these results, and for meta-analytic purposes.

3. In the decoding analysis (Figure 4), the authors test the performance of their decoder using a simple one-sample t-test against chance level performance. While this method was widely used in the early MVPA literature, it is considered overly liberal as the distribution of accuracy data from decoders do not meet the assumptions of the t-test against chance (see Stelzer et al., 2013, Neuroimage; Allefeld et al., 2016, Neuroimage). Please look to these papers and consider adopting a statistical approach that is more in line with known issues in analyzing these decoding accuracy data (e.g. non-parametric permutation tests).

4. In the discussion, the authors claim that "Our results from MVPA analysis revealed that the thalamus preferentially encodes contextual information, but not other task-relevant information within the hierarchical task structure." I am afraid that this reads a bit like accepting the null finding that other task-relevant information is not encoded in the thalamus. This may not be the authors' intent, but I suggest they reword this to make it clear that they simply did not find evidence for such information being encoded in the thalamus, which they may have not had sufficient resolution or power to detect with fMRI.

5. It would be helpful if the authors provided more explanation of their null models for the activity flow analysis in the body of the results section.

6. In the methods section, the authors note that for their RSA analysis, "the neural dissimilarity was computed as one minus the spatial Mahalanobis distance." I think this must be a mistake, as this should only be done on correlation or cosine distances which are on a -1 to 1 scale, not on the Mahalanobis distance which is scaled only by the noise covariance.

7. The methods section mentions that no-smoothing was applied to the data. Is that true for all analyses? The univariate evoked responses (i.e. Figure 2) look as though some smoothing may have been applied. Please clarify where, if at all, smoothing was used - and the size of the smoothing kernel.

8. In Figure 7, it would be helpful to have a breakdown of the drop in prediction accuracy for removing voxels in different parts of the thalamus akin to panel e in Figure 5.

9. The supplementary figures start at S4 rather than S1 and do not always correspond with the text. Please double-check and correct.

10. Figure S7 (titled "Voxel-to-voxel BG-cortical model performance") was not clear to me. Are the top panels showing the results for predicting the data from ROIs identified by the RSA analyses for Context, Context x CO, etc.? I am not sure how these then relate to the task switching conditions (EDS, IDS, stay). Please provide more detail in the Figure legend explaining what the colors for each bar represent and what was being predicted.

11. Please consider including individual data points in behavioral results in Figure 1c and d with lines connecting participants so that it is easier to visualize variability. Please also label error bars in the figure caption.

12. Please consider making S5 (noise ceiling metrics) into a table. It is currently very hard to parse as a paragraph.

Reviewer #3: This manuscript explores the role of the thalamus during a hierarchical decision making task involving frontal cognitive control. The authors collected fMRI data from ~60 healthy participants, and investigated how different hierarchical levels of task representation switch recruit the fronto-thalamic network. To do so, they used standard univariate analyses, decoding, as well as a more novel form of functional connectivity analysis, including simulated lesions in the thalamus. The results showed an increased activation in various thalamic nuclei especially when they had to make a rule based decision switch and that these activations were specific to updating at the highest hierarchical level of contextual representations. Finally, they found that these nuclei interact with other parts of the cortex involved in cognitive control, in particular parts of prefrontal cortex which replicate past findings. 

This manuscript is relevant for the journal's scope and is original, it is overall well written and clear, and the results are original and relevant to the field. In general, the studies cited are appropriate, relevant and most of them are recent. The introduction contains all the relevant references and builds up the authors' hypotheses. The methods section provides essential information to understand the analysis performed. Showing cohen's d is good practice to understand the impact of the results. The main results are well discussed. Overall, we found that this article is very solid and a useful addition to the literature on thalamic contributions to cognitive control. We note that we do not feel qualified to comment on the more intricate fMRI analyses. 

Some points that could strength the manuscript

- Clarification on task structure. The methods don't clarify the randomization of the sequence of cues and stimuli. This is important as it controls the sequence of switches, which is at the core of the further analysis. Furthermore, there is no mention of how stay trials are balanced - in particular, there are two equivalent cues that allow for a stay trial to still be a "cue switch" (as in the figure example), but it is not clear whether all stay trials are "cue switch" trials or if some are also "cue stay" trials. Again, this could impact both the behavioral and imaging findings. 

- Behavioral analysis. The analysis is quite minimal, looking only at EDS, IDS vs. stay. Cue stay vs. switch, and decision stay vs. switch are also known factors impacting RTs and accuracy, in ways that may interact with EDS and IDS. It seems important to analyze this in this context. 

- These factors should also be considered within the fMRI analysis, in particular the initial univariate one and the decoding analyses. It is strange that the "decision" factor is considered in the later part of the paper but not the earlier part. It is clearly an important aspect of representations in this task. 

Minor comments

P11 l1: FC matrix should be defined when first comes up, and it would be helpful to briefly say how it's derived to help understand the model within the main text. 

Fig. S1 is missing a legend for the line colors.

---

## [Decision Letter · Decision Letter 2]

24 Oct 2024

Dear Dr Chen,

Thank you for your patience while we considered your revised manuscript "Thalamocortical contributions to hierarchical cognitive control" for consideration as a Research Article at PLOS Biology. Your revised study has now been evaluated by the PLOS Biology editors, the Academic Editor and one of the original reviewers.

In light of the reviews, which you will find at the end of this email, we are pleased to offer you the opportunity to address the remaining points from the reviewer in a revision that we anticipate should not take you very long. We will then assess your revised manuscript and your response to the reviewers' comments with our Academic Editor aiming to avoid further rounds of peer-review, although might need to consult with the reviewers, depending on the nature of the revisions.

After discussing the report with the Academic Editor, we think it would be sufficient if you redid the analysis as suggested by the reviewer, and if the results don't change much, include a statement in the manuscript that similar results are obtained when using a more established correction procedure. 

**IMPORTANT - SUBMITTING YOUR REVISION**

*Resubmission Checklist*

*Published Peer Review*

*PLOS Data Policy*

*Blot and Gel Data Policy*

Sincerely,

Christian

Christian Schnell, PhD

Senior Editor

PLOS Biology

cschnell@plos.org

REVIEWS:

Reviewer #2: The authors have addressed most of my concerns in the revision of this manuscript. However, I have one remaining concern:

In their revision, the authors clarified the thresholding procedure that they used for the univariate analyses in Figure 2. Namely, they did not perform any smoothing of the data and used a peak-threshold of P < 0.05 and cluster extent threshold of P < 0.05 to correct for multiple comparisons. This is an unusual procedure, as P < 0.05 is a very low peak threshold and could lead to spurious results. However, in absence of any smoothing, this might be acceptable given that the likelihood of detecting a significant cluster will be reduced on unsmoothed data. As this is not a common practice, I am not sure how valid this approach is without simulations to demonstrate its validity. I think it would be helpful if the authors could do this analysis either using threshold free-cluster enhancement or smoothing the data and then applying the standard peak threshold (P < 0.001) before correction for multiple comparisons. I would feel more satisfied with the results if the pattern of results for this univariate analysis is not changed when using more typical methods.

---

## [Editor Report · Decision Letter 3]

8 Nov 2024

Dear Dr Chen,

Thank you for your patience while we considered your revised manuscript "Thalamocortical contributions to hierarchical cognitive control" for publication as a Research Article at PLOS Biology. This revised version of your manuscript has been evaluated by the PLOS Biology editors and the Academic Editor.

Based on our Academic Editor's assessment of your revision, we are likely to accept this manuscript for publication, provided you satisfactorily address the following data and other policy-related requests:

* We would like to suggest a different title to improve accessibility: The thalamus encodes and updates context representations during hierarchical cognitive control

* Please include information in the Methods section whether the study has been conducted according to the principles expressed in the Declaration of Helsinki.

* Please note that per journal policy, we do not allow the mention of data that is not publicly available or contained within this manuscript. Therefore, please provide a figure presenting the results of your additional analyses "using an 8mm smoothing kernel, voxel-level7 threshold of p <0.001, and cluster threshold of p <0.05 (cluster size of 42 voxels)". This can be provided as a Supplementary Figure. 

* DATA POLICY:

Regardless of the method selected, please ensure that you provide the individual numerical values that underlie the summary data displayed in the following figure panels as they are essential for readers to assess your analysis and to reproduce it: 1CD, 5BE, 6D, 7B, 8D, S1ABC, S6B and S9AB

* CODE POLICY

We expect to receive your revised manuscript within two weeks. 

*Published Peer Review History*

*Press*

Sincerely,

Christian

Christian Schnell, PhD 

Senior Editor

cschnell@plos.org

PLOS Biology

---

## [Editor Report · Decision Letter 4]

13 Nov 2024

Dear Dr Chen,

Thank you for the submission of your revised Research Article "The thalamus encodes and updates context representations during hierarchical cognitive control" for publication in PLOS Biology. On behalf of my colleagues and the Academic Editor, Thorsten Kahnt, I am pleased to say that we can in principle accept your manuscript for publication, provided you address any remaining formatting and reporting issues. These will be detailed in an email you should receive within 2-3 business days from our colleagues in the journal operations team; no action is required from you until then. Please note that we will not be able to formally accept your manuscript and schedule it for publication until you have completed any requested changes.

PRESS

Sincerely, 

Christian

Christian Schnell, PhD

Senior Editor

PLOS Biology

cschnell@plos.org